# V2a interneuron diversity tailors spinal circuit organization to control the vigor of locomotor movements

Jianren Song[1,2], Elin Dahlberg[2] & Abdeljabbar El Manira[2]

Locomotion is a complex motor task generated by spinal circuits driving motoneurons in a precise sequence to control the timing and vigor of movements, but the underlying circuit logic remains to be understood. Here we reveal, in adult zebrafish, how the diversity and selective distribution of two V2a interneuron types within the locomotor network transform commands into an appropriate, task-dependent circuit organization. Bursting-type V2a interneurons with unidirectional axons predominantly target distal dendrites of slow motoneurons to provide potent, non-linear excitation involving NMDA-dependent potentiation. A second type, non-bursting V2a interneurons with bidirectional axons, predominantly target somata of fast motoneurons, providing weaker, non-potentiating excitation. Together, this ensures the rapid, first-order recruitment of the slow circuit, while reserving the fast circuit for highly salient stimuli involving synchronous inputs. Our results thus identify how interneuron diversity is captured and transformed into a parsimonious task-specific circuit design controlling the vigor of locomotion.

[1] Center of Translational Medicine, Tongji Hospital, Tongji University School of Medicine, Shanghai 200065, China. [2] Department of Neuroscience, Karolinska Institute, 171 77 Stockholm, Sweden. Correspondence and requests for materials should be addressed to J.S. (email: song.jianren@tongji.edu.cn) or to A.E.M. (email: abdel.elmanira@ki.se)

Our ability to move shapes the way we interact with the outside world. All organisms are endowed with a versatile movement repertoire that can be deployed selectively to achieve intended goals, ranging from manipulating a delicate object to fleeing from a threat[1,2]. Therefore, all motor actions not only need to be planned and initiated, but also precisely executed in terms of both timing and vigor[3–8]. The execution of all intended motor actions ultimately relies on a precise control over the temporal activation sequence of motoneurons, from slow to fast ones. However, it has proven difficult to resolve the circuit rules governing the final step in the transformation of motor commands into the sequential activation of motoneurons for the control of movement vigor.

Most motor commands from the brain are first processed by spinal circuits that form an active interface before they are conveyed to motoneurons to produce movements, such as those underlying locomotion. Neural circuits in the spinal cord, termed central pattern generators (CPGs), generate the locomotor rhythm with its embedded coordination[3,5,9–13]. Locomotor movements are not stereotyped—their vigor (speed and amplitude) varies continuously depending on changes in the outside world. The role of several cardinal interneuron classes in controlling different attributes of locomotion, such as excitatory drive and left–right coordination, has been clarified[3–5,8,14–16], but precisely how the versatility of movement vigor is generated in the spinal cord remains poorly understood. The dominant view is that versatility in the vigor of movements arises from differences in the intrinsic properties of motoneurons[17–23]. This flexibility could also emerge from the functional diversity of the excitatory interneurons that drive motoneurons[24]. This study focuses on an untested alternative—that the control of the vigor of locomotor movements is implemented by a combination of diverse intrinsic properties of excitatory interneurons, combined with specialized connectivity patterns, temporal dynamics, and synaptic integration.

The zebrafish shares common molecular mechanisms for spinal cord development with mammals and other vertebrate species[3,4,25–27]. Yet, the experimental amenability of zebrafish allows a detailed circuit analysis to be performed with consideration to the diversity of CPG neurons. Within the locomotor network, CPG excitation is conveyed by the *Chx10*-expressing V2a interneurons, which when selectively activated using optogenetics elicits coordinated locomotion[28–31]. The spinal circuits in juvenile/adult zebrafish are modularly organized, with the excitatory V2a interneuron population consisting of three subclasses that selectively connect to slow, intermediate or fast motoneurons, forming three modules that can be successively recruited to increase the locomotor speed[14,24]. However, at present, these modules are viewed as reiterations of a canonical circuit that obeys the same construction scheme, without consideration of the extent, and functional consequences, of diversity within the V2a interneuron population.

Here we have uncovered a previously unappreciated degree of diversity of the V2a interneuron population and demonstrate how this is translated into a module-specific circuit design supplemented by an optimized synaptic integration mode that selectively channels an attuned excitatory drive. Our results show that within each speed module there are two types of V2a interneurons with fundamentally different firing patterns, axonal projections, spatial locations of their synapses onto motoneurons, and synaptic integration modes. The two V2a interneurons types are differentially distributed across the three speed modules, tailoring the circuit design and synaptic integration mode within each module according to the vigor of movement they underlie. Bursting-type V2a interneurons with unidirectional axons target the distal dendrites of predominantly slow motoneurons, and provide a potent non-linear excitation involving NMDA-dependent potentiation. A second type, non-bursting V2a interneurons with bidirectional axons, target the soma of predominantly fast motoneurons, and provide weaker, non-potentiating excitation. Thus, these findings provide fundamental insights into how neuronal diversity is translated into circuit-specific assembly with an attuned synaptic integration that is indispensable for controlling changes in the vigor of locomotor movements.

## Results

**Modularity and unequal distribution of V2a interneurons and motoneurons.** We first sought to examine whether the anatomical distribution and convergence of V2a interneurons onto the motoneuron population is sufficient to account for the progressive motoneuron pool recruitment during increased swimming vigor. To this end, we utilized a transgenic zebrafish line selectively expressing GFP in V2a interneurons[29,32]. We first compared the relative numbers of V2a interneurons and motoneurons, with the latter labeled retrogradely by injecting a rhodamine dextran dye into axial muscles of adult zebrafish[33,34]. None of the retrogradely labeled motoneurons expressed GFP, which was restricted only to V2a interneurons[24,29,31], while practically all GFP-expressing V2a interneurons were immunoreactive for *Chx10* (Supplementary Fig. 1). The number of motoneurons ranged between 36 and 43 per hemi-segment, whereas V2a interneurons ranged between 16 and 23 per hemi-segment ($n = 13$) (Fig. 1a, b). The number of V2a interneurons per hemi-segment is similar to what has been previously reported in zebrafish larvae[35]. These results show that in adult zebrafish, motoneurons outnumber V2a interneurons by approximately a two-to-one ratio.

The unequal motoneuron/V2a interneuron distribution could be compensated for by a high degree of convergence. Therefore, we estimated the number of V2a interneurons converging onto different motoneuron pools. A large proportion of V2a interneurons and motoneurons are connected via mixed chemical and electrical synapses (65% of the connections), while a small proportion (35% of the connections) are only connected via chemical synapses[36]. This enables retrograde diffusion of dye via gap junctions (dye-coupling), and hence allows for an estimation of the degree of convergence from V2a interneurons onto motoneurons. To this end, neurobiotin was injected selectively into either slow muscle, fast muscle, or both slow and intermediate muscles, and the number of dye-coupled V2a interneurons was estimated. Motoneurons in each of the three pools received converging inputs from 2 to 4 V2a interneurons in the same segment ($n = 14$), and 1 to 4 V2a interneurons from the neighboring 4 segments ($n = 14$) (Fig. 1c–h). The convergence of several V2a interneurons onto a single motoneuron was further confirmed by intracellular injection of neurobiotin into single motoneurons. This resulted in dye-coupling of 2–3 segmental, and 1–2 intersegmental, V2a interneurons from a slow, an intermediate or a fast motoneuron ($n = 5$ for each motoneuron type; Supplementary Fig. 2). This analysis only provides an underestimation of the number of V2a interneurons converging onto motoneurons because not all the connections have an electrical component. Nonetheless, these results indicate that there is no bias in the probability of connections between V2a interneurons and motoneurons of the slow, intermediate and fast circuit modules.

These findings show a sparse anatomical distribution and synaptic convergence of V2a interneurons onto motoneurons. This anatomical disparity would seem unfavorable for an accurate and progressive recruitment of the different motoneuron pools

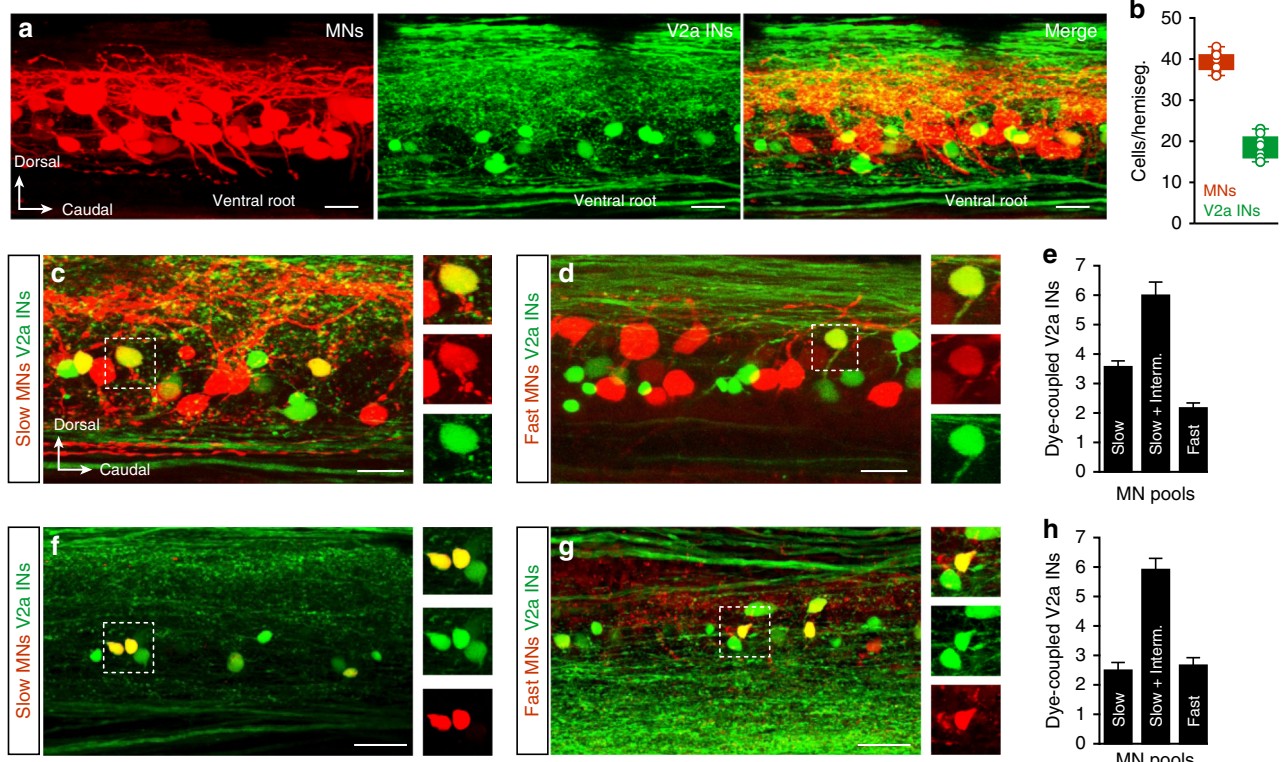

**Fig. 1** Distribution of motoneurons and V2a interneurons. **a** Lateral view of the spinal cord with retrogradely labeled motoneurons by injection of Rhodamine-dextran into the muscle (left), GFP-expressing V2a interneurons (middle) and the merge of the two (right). **b** There were more motoneurons than V2a interneurons in each hemi-segment of the spinal cord ($n = 13$ animals, the boxes are bound by the 25th and 75th percentiles, whiskers extend from min. to max.). **c**–**e** Number of segmental V2a interneurons dye-coupled to slow, intermediate and fast motoneurons ($n = 14$ animals, error bars in the graph reflect the s.e.m.). **f**–**h** Number of intersegmental V2a interneurons in the 4 adjacent segments (2 rostral and 2 caudal) dye-coupled to slow, intermediate and fast motoneuron pools ($n = 14$ animals, error bars in the graph reflect the s.e.m.). Scale bars, 20 μm

necessary for controlling swimming movement, and therefore further mechanisms must be involved.

**Functional diversity of V2a interneurons and their module-specific connectivity.** We next determined the extent of V2a interneuron diversity, which could potentially endow them with mechanisms to amplify their excitatory drive to motoneurons in a module-specific manner, and hence compensate for their sparse anatomical convergence. To this end, we examined how V2a interneurons within each of the three modules differ in their firing properties, and in the excitatory synaptic drive they provide to their target motoneurons. Remarkably, this analysis revealed a previously undescribed layer of V2a interneuron diversity. Within each speed module, two functional V2a interneuron types emerged that display distinct firing patterns and synaptic strengths. One type, bursting V2a, fired with rhythmic bursts of action potentials in response to injected current steps, while a second type, non-bursting V2a, fired regularly with mild adaptation (Fig. 2a–c). The bursting-type V2a interneurons always elicited monosynaptic excitatory postsynaptic potentials (EPSPs) with significantly larger amplitude than the non-bursting-type, irrespective of their modular identity ($f_{1,90} = 73.25$; $p < 0.0001$; two-way repeated-measures ANOVA; Fig. 2a–d). Moreover, the amplitude of the EPSPs elicited by the bursting-type V2a interneurons was larger in the slow module compared to the intermediate and fast modules ($f_{2,90} = 4.99$; $p = 0.0087$; two-way repeated-measures ANOVA; Fig. 2d). In contrast, the amplitude of the EPSPs elicited by the non-bursting-type V2a interneurons was similar across the three modules ($p > 0.05$; two-

way repeated-measures ANOVA; Fig. 2d). There was no correlation between the motoneuron input resistance and the EPSP amplitude induced by bursting- (Fig. 2e) or non-bursting-type V2a interneurons (Fig. 2f).

In addition to their physiological and synaptic properties, the bursting-type and non-bursting-type V2a interneurons displayed distinct morphologies with respect to their axonal projections and the spatial location of their synapses onto motoneurons. The bursting-type V2a interneurons were unidirectional, with their axon projecting caudally. On the other hand, the non-bursting-type V2a interneurons were bifurcating, with bidirectional axonal projections in both the rostral and caudal directions (Fig. 3a). The bursting-type V2a interneurons targeted the distal dendrites of postsynaptic motoneurons, while the non-bursting-type interneurons targeted the somatic region of motoneurons (Fig. 3b). There was also a differential connectivity pattern between bursting-type and non-bursting-type V2a interneurons and motoneurons of each speed module (Fig. 3c, d). Slow motoneurons preferentially received dendritic synaptic inputs from bursting-type V2a interneurons (82% of connections onto dendrites) ($n = 12$; Fig. 3c, d), whereas fast motoneurons received synaptic inputs mostly on their somatic region from non-bursting-type V2a interneurons (75% of connections onto soma) ($n = 13$; Fig. 3c, d). The intermediate motoneurons received inputs from both bursting- and non-bursting-type V2a interneurons that were uniformly distributed on their distal dendrites and somatic region (54% of the connections on the dendrites and 46% on the soma region) ($n = 13$; Fig. 3c, d).

These results show that V2a interneurons of each of the three modules could be further segregated into two functional types

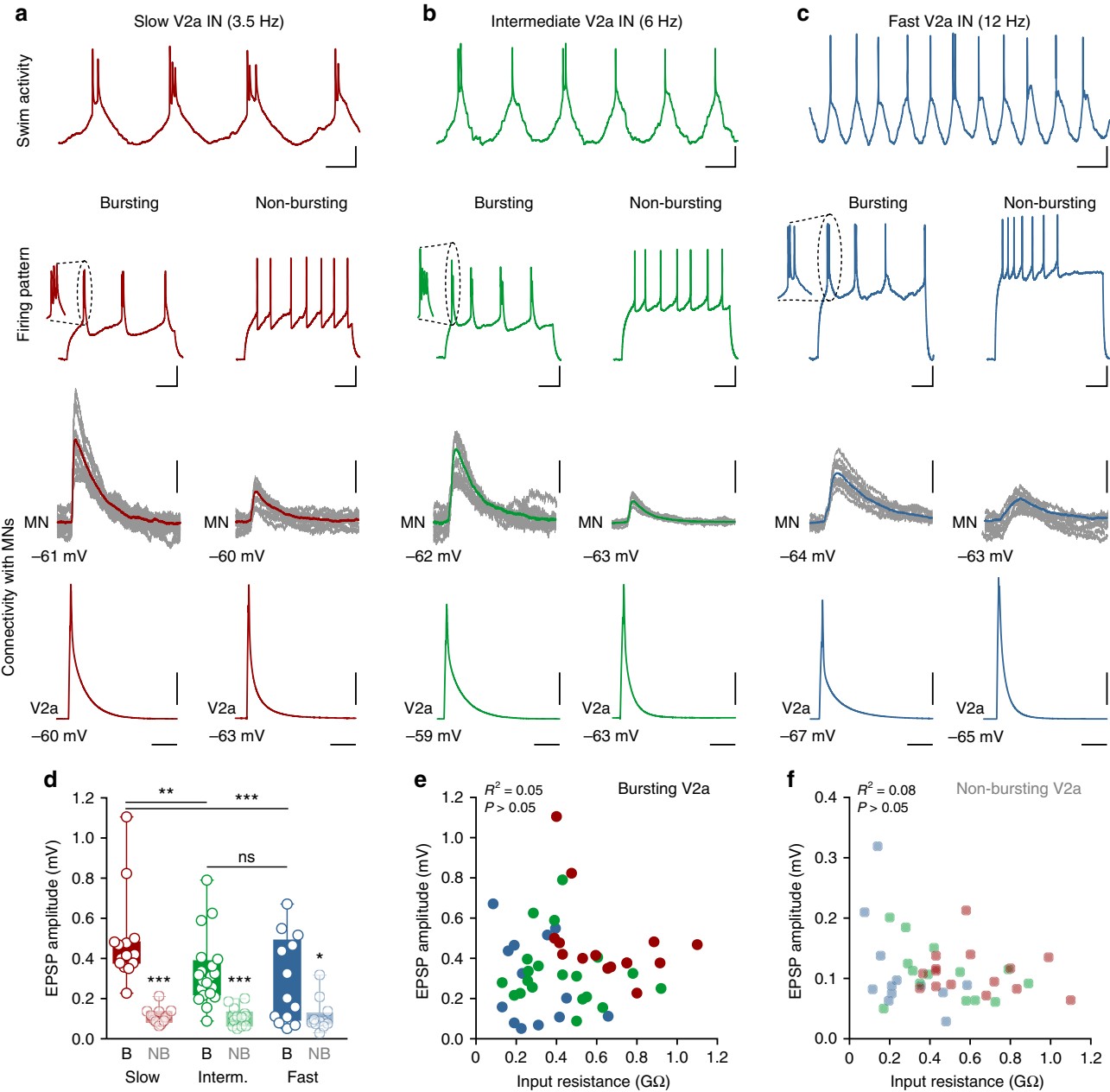

**Fig. 2** Functional diversity of V2a interneurons and their synaptic drive to motoneurons. **a** Upper panel: swimming activity of a slow V2a interneuron. Middle panel: V2a interneurons of the slow circuit module displayed either bursting (left) or non-bursting (right) firing pattern in response to depolarizing current pulses. Lower panel: slow bursting-type V2a interneurons (left) always produced much larger EPSPs than non-bursting-type (right) in slow motoneurons. **b** Upper panel: swimming activity of an intermediate V2a interneuron. Middle panel: intermediate V2a interneurons consist of bursting- (left) and non-bursting types (right). Lower panel: intermediate bursting-type V2a interneurons (left) produced larger EPSPs than non-bursting-type (right) in intermediate motoneurons. **c** Upper panel: swimming activity of a fast V2a interneuron. Middle panel: V2a interneurons of the fast module also consist of bursting (left) and non-bursting types (right). Lower panel: fast bursting-type V2a interneurons (left) produced larger EPSPs than non-bursting-type (right) in fast motoneurons. **d** The amplitude of the monosynaptic EPSPs induced by V2a interneurons in the target motoneurons of the same module was always larger for the bursting-type (B) than non-bursting-type (NB) interneurons across modules (Interaction: $F_{2,90} = 5.63$; $P = 0.005$; two-way repeated-measures ANOVA; bursting vs non-bursting: $F_{1,90} = 73.25$; $P < 0.0001$; two-way repeated-measures ANOVA; $n = 15$ pairs of the slow module; $n = 20$ pairs of the intermediate module; and $n = 13$ of the fast module; the boxes are bound by the 25th and 75th percentiles, whiskers extend from min. to max.). **e**, **f** There was no correlation between the motoneuron input resistance and the EPSP amplitude induced by bursting-type (**e**) and non-bursting-type (**f**) V2a interneurons ($P > 0.05$; Pearson's correlation). Scale bars, 100 ms, 10 mV (swim and firing pattern); 20 ms, 10 mV, 0.2 mV (connectivity)

that are connected with motoneurons in a module-specific manner, such that motoneurons within the slow speed module preferentially receive inputs from bursting-type V2a interneurons, whilst those in the fast module receive inputs mainly from non-bursting-type V2a interneurons.

**The diversity of V2a interneurons is translated into module-specific synaptic integration**. Next, we examined if the differential assembly between the two V2a interneuron types and motoneurons endows each of the three modules with a tailored synaptic integration mode, that amplifies the excitatory drive in a

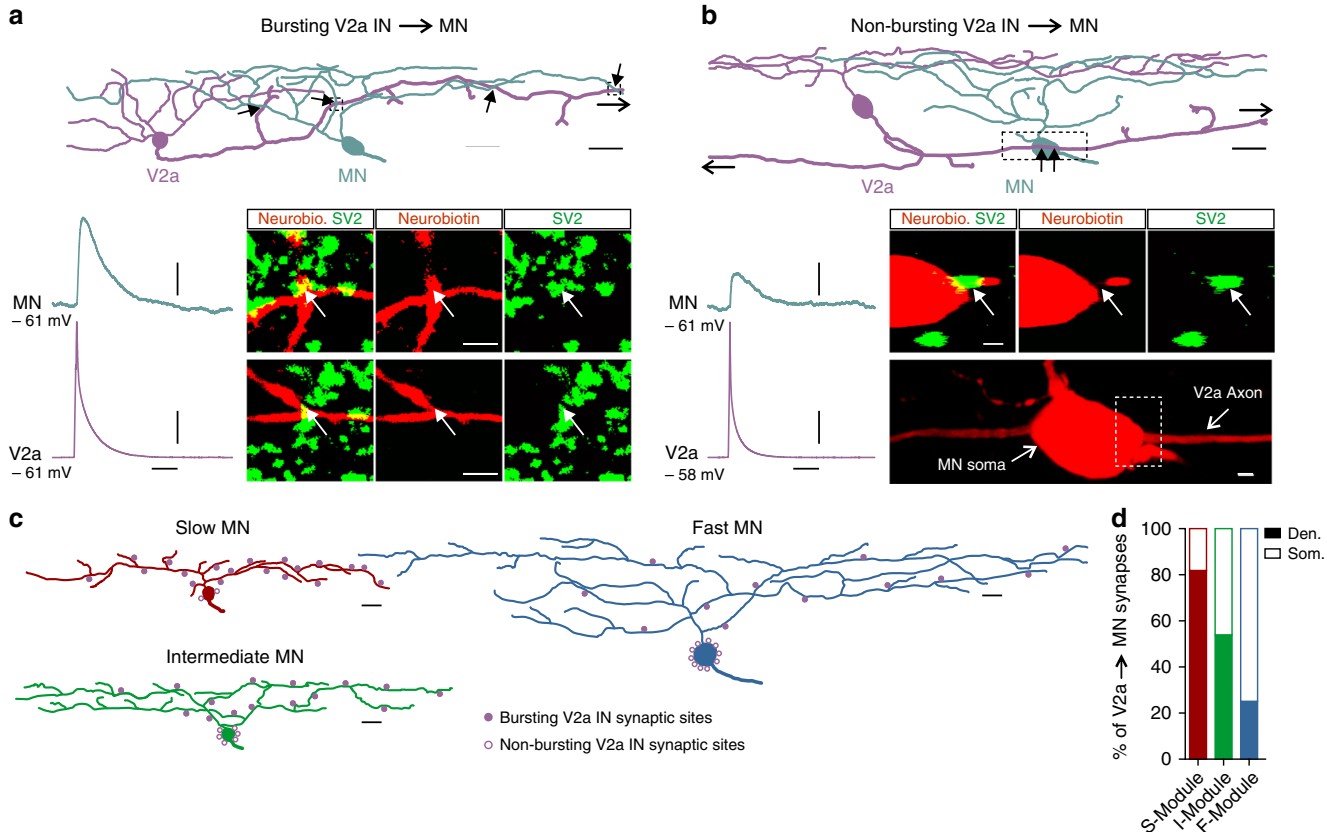

**Fig. 3** Distinct axonal projections of the two V2a interneuron types. **a** The bursting-type V2a interneurons had unidirectional descending axons that target the dendrites of the postsynaptic motoneurons where they produce strong excitatory drive. The filled-head black arrows indicate the sites of synaptic contacts between the V2a interneuron axon and the motoneuron dendrites. The open-head arrow reflects the fact that the axon project further than shown in this reconstruction. The dashed boxes indicate the regions that are enlarged in the lower panel showing optical sections of the sites of synaptic contacts between the V2a interneuron axon and the motoneuron dendrite that also co-localized the synaptic vesicle protein SV2 (white arrow). **b** The non-bursting-type V2a interneurons had bi-directional axons projecting in the caudal and rostral directions. They target the soma region of the postsynaptic motoneurons where they produce weak excitatory drive. The filled-head black arrows indicate the site of contacts between the axon of the V2a interneuron and motoneuron soma. The open-head arrows reflect the fact that the axon project further than shown in this reconstruction both in the rostral and caudal directions. The dashed box corresponds to the enlarged region in the lower panel showing an optical section where the sites of synaptic contacts between the V2a axon and the motoneuron soma as well as the localization of the synaptic vesicle protein SV2 (white arrows). **c** Reconstructions showing the distribution of synaptic contacts on the dendrites and soma region of the slow, intermediate and fast motoneurons from bursting-type (filled circles) and non-bursting-type (open circles) V2a interneurons. **d** Percentage of dendritic (filled part of the bar) versus somatic (open part of the bar) synaptic contacts onto motoneurons of the slow, intermediate and fast modules. ($n = 11$ slow motoneurons; $n = 13$ intermediate motoneurons; and $n = 11$ fast motoneurons). Scale bars, 10 µm, 1 µm, 20 ms, 10 mV, 0.2 mV

module-specific manner in accordance with their function. Two different modes of synaptic integration, a supra-linear (non-linear) and a linear one, were revealed. Synaptic connections involving bursting-type V2a interneurons displayed a powerful, non-linear integration that amplifies the EPSPs elicited in their target motoneurons (Fig. 4). A V2a interneuron burst produced significant larger EPSPs than those elicited by single action potentials (Fig. 4a–c). Notably, the maximum amplitude of the burst-induced EPSPs was graded as a function of the modular identity of the V2a-motoneuron pairs (Fig. 4d). There was a strong correlation between the swimming frequency at which the bursting-type V2a interneurons became recruited and the maximum amplitude of the burst-induced EPSPs in the corresponding motoneuron (Fig. 4d). The amplitude of the EPSPs induced by bursting-type V2a interneurons increased as a function of the number of action potentials during a burst (1–5 action potentials) induced by depolarizing current steps. The resulting EPSPs in the target motoneurons displayed a strong short-term facilitation producing a supra-linear summation,

with the maximum amplitude induced by bursts of 4 to 5 action potentials (Fig. 4e). Furthermore, the magnitude of the non-linear summation was organized in a module-specific fashion, being highest in V2a interneuron-motoneuron synapses of the slow module, and lowest in pairs of the fast module (Fig. 4e). Overall, the strength of synaptic transmission and the non-linear amplification of the EPSPs during a burst are precisely attuned in a module-specific manner. Thus, not only is there a higher proportion of bursting-type V2a interneurons in the slow speed module, but also the potentiation of their inputs to slow motoneurons is strongest compared to the bursting-type V2a inputs to motoneurons in the intermediate and fast modules.

We next wanted to test whether this potentiation, and the module-specific gradient of its magnitude, can be accounted for by NMDA-mediated, dendritic non-linearity in the motoneurons. Bursting-type V2a interneurons were depolarized using current injections with increasing duration to elicit bursts with 1 to 5 action potentials, and the role of NMDA receptors was tested

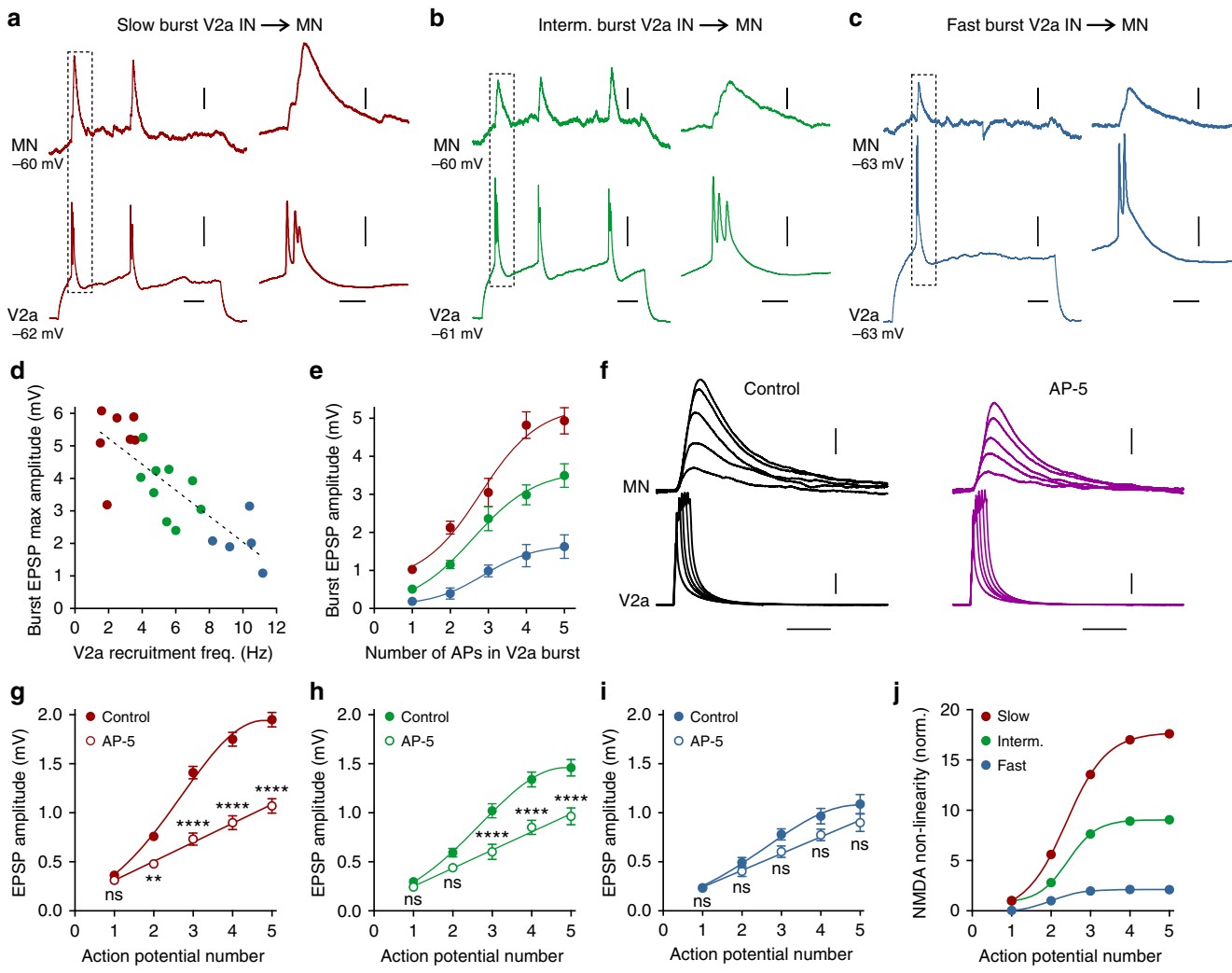

**Fig. 4** NMDA-dependent short-term potentiation of bursting-type V2a interneuron synaptic transmission. **a–c** Single bursts of action potentials in V2a interneurons of the slow, intermediate and fast modules produced a large monosynaptic EPSPs in the target motoneurons. **d** The maximum amplitude of the burst-induced EPSPs was graded according to the order of recruitment of the V2a interneurons during swimming. Each point in the graph reflects a single measurement from different V2a interneuron-motoneuron pairs. **e** The EPSPs elicited by bursting V2a interneurons showed a strong non-linear summation as a function of the number of action potentials in a burst. The magnitude of the short-term potentiation produced by the bursting-type V2a interneurons conformed to their modular identity ($n = 6$ pairs of slow module; $n = 6$ pairs of intermediate module; and n = 6 of fast module; error bars in the graph reflect the s.e.m.). **f** A similar non-linear summation was produced by current injections into the bursting-type V2a interneurons that mimicked a normal burst with increasing number of action potentials. The short-term potentiation was blocked by the NMDA receptor antagonist AP-5. **g–i** Summary of the results of experiments shown in **f**. Blockade of NMDA receptors switched the integration mode from non-linear to linear summation ($n = 6$ pairs of each module; error bars in the graph reflect the s.e.m.; interaction slow: $F_{4,50} = 26.20$, $P < 0.0001$; intermediate: $F_{4,50} = 6.14$, $P = 0.0004$; fast: $F_{4,50} = 0.99$, $P = 0.4197$; two-way repeated-measures ANOVA: ****$P < 0.0001$; ***$P < 0.001$; **$P < 0.01$). **j** The magnitude of the NMDA-mediated nonlinearity was largest in the slow module and lowest in the fast module. Data in this graph were obtained by subtracting the mean EPSP amplitude in AP-5 from control. Scale bars, 50 m, 10 ms, 10 mV, 1 mV **a–c**, 20 ms, 10 mV, 0.4 mv **f**

(Fig. 4f). The increase in the number of V2a interneuron action potentials resulted in a supra-linear summation of the EPSPs. Blockade of NMDA receptors suppressed this short-term potentiation and switched the mode of synaptic integration from a supra-linear to a linear summation (Fig. 4f–i) both in connections with mixed synapses (electrical and chemical), and with only chemical synapses (Supplementary Fig. 3a–c). This demonstrates that the observed short-term potentiation relies on dendritic, NMDA-mediated non-linearity in motoneurons. The magnitude of the potentiation was scaled in a module-specific manner, being maximum in V2a interneuron-motoneuron connections of the slow module, and minimum in those of the fast module (Fig. 4j).

In contrast, the non-bursting V2a interneurons relied only on a linear mode of integration that did not involve NMDA receptor activation in motoneurons both in connections with mixed chemical and electrical synapses, and in those with only chemical synapses (Supplementary Figure 3d–f). Non-bursting-type V2a interneurons induced very small EPSPs in the target motoneurons (Fig. 5a) and there was no short-term potentiation as a function of increasing numbers of action potentials in these V2a interneurons (Fig. 5b). The amplitude of the EPSPs in the target motoneurons was similar across all three modules and increased linearly with the number of action potentials of non-bursting-type V2a interneurons (Fig. 5c). Blockade of NMDA receptors produced

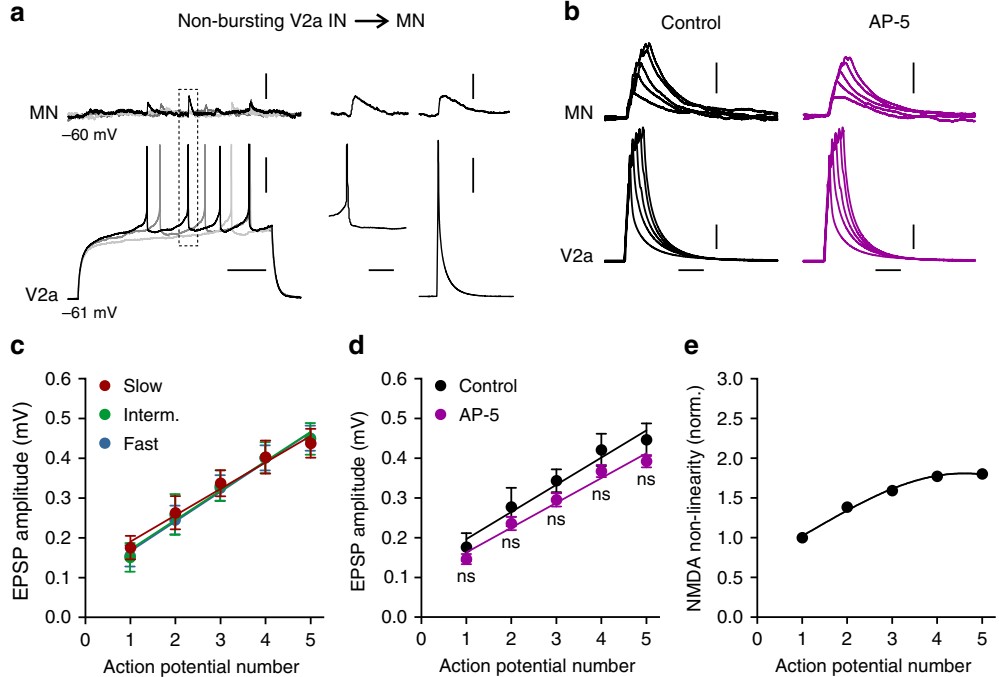

**Fig. 5** Absence of NMDA-dependent short-term potentiation in non-bursting-type V2a interneurons. **a** Repetitive action potentials in a non-bursting-type V2a interneuron elicited small amplitude EPSPs in a target motoneuron, similar to those produced by single action potentials. **b** Stimulation of a non-bursting-type V2a interneuron with current pulses with increasing duration to mimic burst firing did not produce any NMDA-dependent non-linear summation. **c** The EPSP amplitude showed a linear summation as a function of the increased number of action potentials in non-bursting-type V2a interneurons of all modules ($n = 6$ in each module; error bar reflects the s.e.m.). **d** Blockade of NMDA receptors did not significantly decrease the amplitude of the EPSPs (interaction: $F_{4,110} = 0.095$, $P = 0.9837$; two-way repeated-measures ANOVA; $n = 12$; error bars in the graph reflect the s.e.m.). **e** There was no NMDA-dependent amplification of the summation of the EPSPs. Scale bars, 100 ms, 10 ms, 10 mV, 0.2 mV

an almost linear, albeit non-significant, decrease in the EPSP amplitude (Fig. 5d, e).

**NMDA-mediated non-linearity sets the module-specific recruitment of motoneurons**. The above results indicate that bursting- and non-bursting-type V2a interneurons target motoneurons in a module-selective manner, supplemented with either a non-linear NMDA-dependent mode of synaptic integration, or a linear NMDA-independent mode of synaptic integration, respectively. We next tested if these module-specific modes of synaptic integration enable the locomotor CPG to favor the early recruitment of motoneurons of the slow module, and to postpone the recruitment of motoneurons of the fast module. This was examined by specifically blocking NMDA receptors in the recorded motoneurons using the activity-dependent antagonist MK 801, which blocks only open NMDA channels. Blockade of NMDA receptors in individual motoneurons affected their locomotor-related activity in a module-dependent manner. In motoneurons of the slow ($n = 8$) and intermediate ($n = 8$) modules, which are preferentially connected to bursting-type V2a interneurons, MK 801 dramatically altered their activity. It gradually decreased their excitatory drive and ultimately stopped their firing (Fig. 6a–f). In contrast, the activity of motoneurons of the fast module, preferentially connected to non-bursting-type V2a interneurons, was not significantly affected by MK-801 ($n = 8$; Fig. 6g–k). Fast motoneurons usually receive only subthreshold synaptic inputs during normal swimming (Fig. 6g) and are only recruited during a few cycles of fast swimming elicited by stimulation close to the Mauthner cell region ($n = 6$; Fig. 6j). Neither the amplitude of the subthreshold excitatory drive, nor the firing, was affected by selective blockade of NMDA receptors by MK-801 in fast motoneurons (Fig. 6g–k). This indicate that the recruitment

of fast motoneurons during swimming does not require activation of NMDA receptors. These results demonstrate that the diversity of V2a interneurons is translated into different modes of synaptic integration within the locomotor CPG to scale the excitatory drive in a module-specific manner, and hence control the graded order of recruitment of motoneurons, from slow to fast, to mediate an increase in the vigor of locomotor movements.

Altogether, these results link the functional diversity of V2a interneurons to their module-specific assembly with motoneurons, endowing each speed module with a preferential mode of synaptic integration that is tailored to their function.

**Impact of V2a interneuron diversity on the module-specific circuit design**. The three V2a interneuron-motoneuron CPG modules do not conform to a reiterated canonical circuit construction. Rather, they have a module-specific circuit design afforded by the functional diversity of the V2a interneurons. This module-specific circuit construction, with distinct synaptic integration modes, can compensate for the sparse V2a interneurons anatomical distribution (16–23 V2a interneurons per hemi-segment) and their convergence onto motoneurons (convergence of 1–4 segmental or intersegmental V2a interneurons onto a single motoneuron). To determine if this is the case, we estimated the number of converging bursting-type vs non-bursting-type V2a interneurons required to recruit a slow, intermediate and fast motoneuron.

We first calculated the amplitude of depolarization required to reach firing threshold in a motoneuron in each speed module (Supplementary Fig. 4). Next, we estimated the number of converging V2a interneurons required to reach this firing threshold by comparing the following three possibilities: (1) excitation from non-bursting-type V2a interneurons with linear summation,

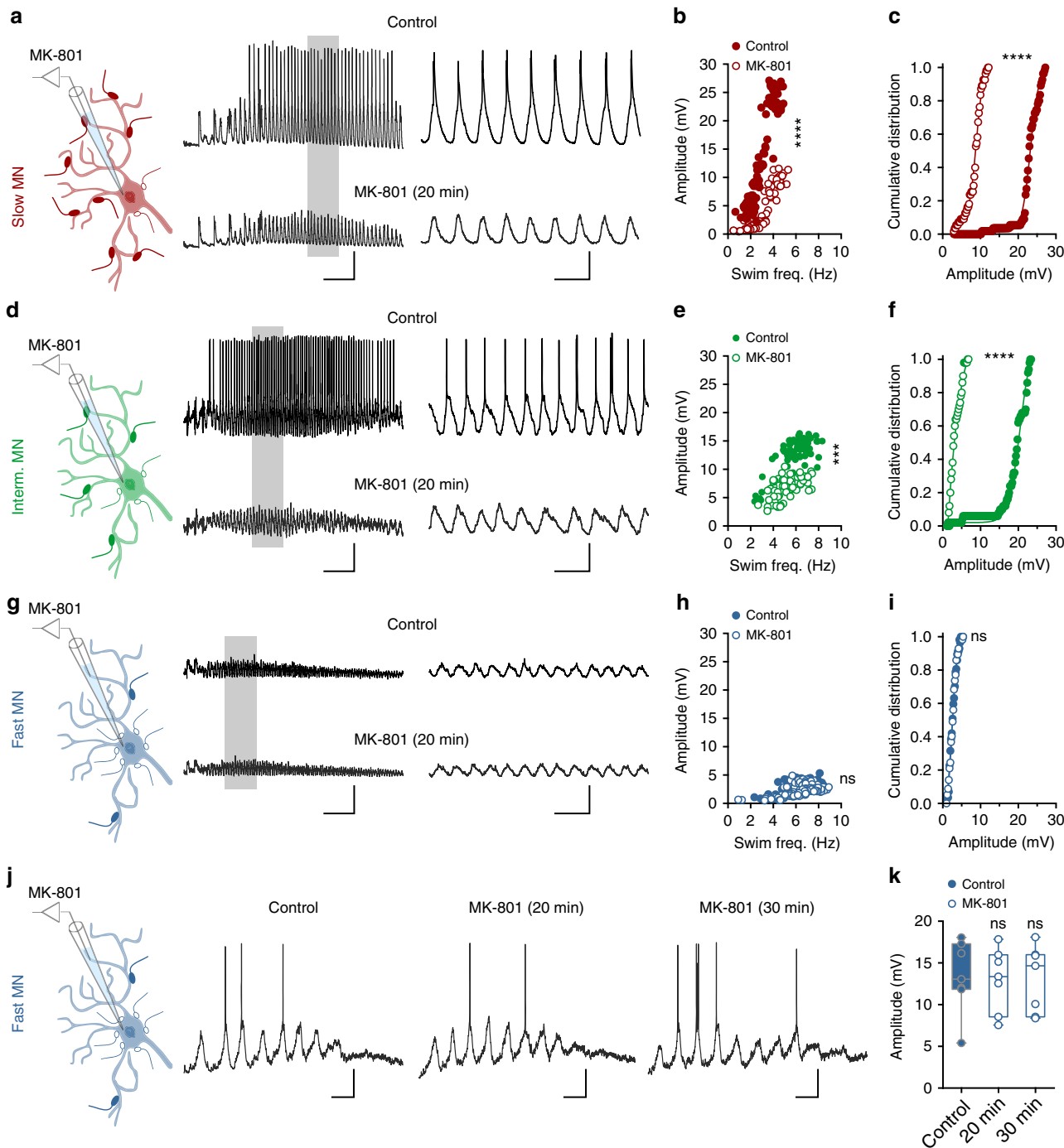

**Fig. 6** NMDA-dependent mode of integration determines motoneuron recruitment order. **a** Selective blockade of NMDA receptors in a slow motoneuron by the intracellular dialysis of MK-801 suppressed the recruitment of these motoneuron during locomotion. **b**, **c** The amplitude of the rhythmic depolarization underlying locomotion in slow motoneurons was decreased by MK-801 ($n = 8$ motoneurons from 8 separate animals; ****$P < 0.0001$; two-tailed Student $t$-test for graph in **b** and Kolmogorov–Smirnov test for graph in **c**). **d** Selective blockade of NMDA receptors in intermediate motoneurons prevented the recruitment of these motoneurons during locomotion. **e**, **f** MK-801 decreased the amplitude of the depolarization in intermediate motoneurons ($n = 8$ motoneurons from 8 separate animals; ****$P < 0.0001$; ***$P < 0.001$; two-tailed Student $t$-test for graph in **e** and Kolmogorov–Smirnov test for graph in **f**). **g** In fast motoneurons, blockade of NMDA receptors by MK-801 had no effect on the amplitude of their subthreshold rhythmic drive. **h**, **i** There was no change in the amplitude of the depolarization in fast motoneurons ($n = 8$ motoneurons from 8 separate animals; $P > 0.05$; two-tailed Student $t$-test for graph in **h** and Kolmogorov–Smirnov test for graph in **i**). **j**, **k** Similarly, MK-801 had no effect on the firing of fast motoneurons during locomotion ($n = 6$ motoneurons from 6 separate animals; $P > 0.05$; two-tailed Student $t$-test; the boxes are bound by the 25th and 75th percentiles, whiskers extend from min. to max.). Scale bars, 2 s, 400 ms, 10 mV **a–g**, 100 ms, 10 mV **j**

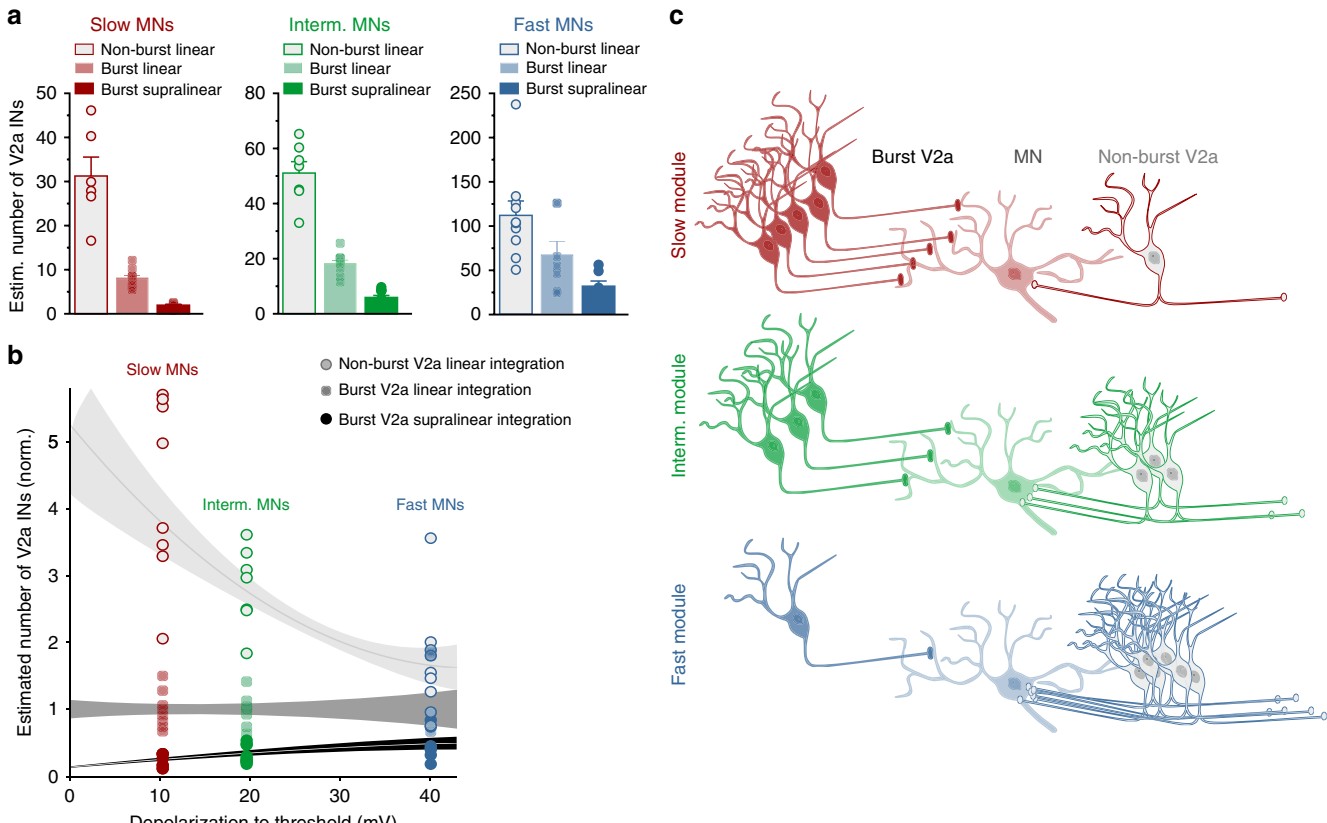

**Fig. 7** Module-specific convergence pattern of bursting- and non-bursting-type V2a interneurons. **a** The estimated number of converging V2a interneurons required to recruit slow, intermediate or fast motoneurons. For the bursting-type V2a interneuron this was calculated based either on supra-linear (non-linear) or linear summation. For non-bursting, this estimation was based only on linear integration. The data represent means and error bars reflect s.e.m. **b** Impact of the synaptic integration modes of bursting- versus non-bursting-type V2a interneurons on the degree of convergence of V2a interneurons required to drive the recruitment of motoneurons in the slow, intermediate and fast modules. The degree of convergence was calculated by normalizing to the estimated number of V2a interneurons in **a** based solely on linear summation of EPSPs from bursting V2a interneurons. The curves reflect the exponential fit with s.e.m. connecting data from slow, intermediated and fast motoneurons. Horizontal curve (dark gray): bursting-type V2a with linear summation used as reference, upper curve (light gray): non-bursting-type V2a with linear summation, and lower curve (black): bursting-type V2a with non-linear summation. **c** Module-specific circuit organization. The slow module relies on excitatory drive with strong non-linearity from bursting-type V2a interneurons, while the fast module relies on weak excitation with only linear summation from non-bursting V2a interneurons. The intermediate motoneurons rely on excitatory drive with a hybrid non-linear and linear integration mode from bursting- and non-bursting-type V2a interneurons, respectively

(2) excitation from bursting-type V2a interneurons with only linear summation, or (3) excitation from bursting-type V2a interneurons with non-linear, NMDA-dependent summation. For the linear summation, the number of single EPSPs necessary to bring the different motoneurons to firing threshold was calculated using the EPSP amplitude induced by single action potentials from bursting-type and non-bursting-type V2a interneurons, respectively (see Fig. 2d). Then, the number of converging V2a interneurons was estimated by dividing this calculated number of EPSPs by the average number of action potentials fired during each locomotor cycle (on average 3 per locomotor cycle)[31]. For the non-linear summation, however, the estimated number of converging V2a interneurons corresponded to the number of burst-induced compound EPSPs (see Fig. 4d) necessary to depolarize the different motoneurons to firing threshold.

The least favorable solution was a circuit construction involving only non-bursting-type V2a interneurons, which provide only a weak excitation associated with a linear synaptic integration. In this configuration, a convergence of 31 V2a interneurons was necessary for the recruitment of a slow motoneuron, 51 V2a interneurons for the recruitment of an intermediate motoneurons,

and 112 interneurons for the recruitment of a fast motoneuron (Fig. 7a). Thus, for this configuration the minimum number of converging V2a interneurons required for recruitment of motoneurons far exceeded the real number of V2a interneurons in the circuit, that ranges between 16 and 23 V2a interneurons in each hemi-segment of the spinal cord (see Fig. 1b). In contrast, the strong excitation provided by the bursting-type V2a interneurons even with only linear summation resulted in a large decrease in the number of converging V2a interneurons required for recruitment of motoneurons in a module-specific manner (Fig. 7a). In this configuration, 8 V2a interneurons were necessary for the recruitment of a slow motoneuron, 18 V2a interneurons for the recruitment of an intermediate motoneuron, and 66 V2a interneurons for the recruitment of a fast motoneuron. The most favorable configuration is a circuit construction with bursting-type V2a interneurons displaying non-linear burst-induced synaptic integration. In this configuration, only 2 V2a interneurons were sufficient to recruit a slow motoneuron, 6 V2a interneurons for the recruitment of an intermediate motoneuron, and 31 V2a interneurons for the recruitment of a fast motoneuron (Fig. 7a).

When comparing the different circuit configurations, there was a major scaling (increase and decrease) in the number of V2a interneurons required to drive the different motoneuron pools to firing threshold. The magnitude of this scaling was organized in a module-specific manner and was influenced both by the functional diversity of V2a interneurons (non-bursting-type vs bursting-type) and the mode of synaptic integration (linear vs non-linear integration) (Fig. 7b). For each module, the scaling factor (degree of convergence) was calculated by normalizing to the predicted number of V2a interneurons based solely on linear summation of EPSPs from bursting V2a interneurons. The degree of convergence was scaled-up (increased) by the type of V2a interneurons (non-bursting vs bursting). Conversely, it was scaled-down (decreased) by the mode of synaptic integration (linear vs non-linear summation). The magnitude of the scaling in both directions was graded in a module-specific manner, being highest in the slow module and lowest in the fast one (Fig. 7b). For the slow module, the V2a interneuron convergence increased from 8 to 31 and decreased from 8 to 2 (a 4-fold scaling factor). For the intermediate modules, the convergence increased from 18 to 51 and decreased from 18 to 6 (a 3-fold scaling factor). Finally, for fast module the convergence increased from 66 to 122 and decreased from 66 to 31 (a 2-fold scaling factor) (Fig. 7b).

These results demonstrate how the functional diversity of V2a interneurons, their connectivity pattern, and their modes of synaptic integration, endow the locomotor CPG with a module-specific parsimonious circuit design that compensates for their sparse anatomical distribution in the network (Fig. 7c).

## Discussion

The ability to generate and refine movements to achieve an intended outcome is a critical function of the nervous system. The execution of the appropriate action relies on sophisticated circuit and synaptic computations that are ultimately reflected in the functional diversity of the neuronal populations involved. However, it has proven difficult to resolve the nature of these computations, how they emerge from the diversity of the circuit's constituent neurons, and how they ultimately produce movements with the intended timing, force and speed for a particular movement. Here we have addressed these questions in the context of a spinal locomotor circuit. We show that the diversity of excitatory V2a interneurons, within a tailored, module-specific circuit configuration, endows the CPG network with a precise control over the vigor of locomotor movements through mechanisms involving NMDA-dependent non-linearity.

V2a interneurons generate the locomotor rhythm and channel the excitatory drive from the locomotor CPG to motoneurons[28–30,32,37]. They consist of three sub-classes, selectively connected to motoneurons innervating slow, intermediate or fast muscles, forming three separate modules underlying the increase in locomotor speed[24,31]. Our findings now show a previously unappreciated level of diversity of the V2a interneuron population, and illustrate how it endows each CPG speed module with distinct circuit assembly and modes of integration that are functionally linked to their range of operation during locomotion. Across these modules, two main types of V2a interneurons emerge, exhibiting distinct electrophysiological signatures, axonal projections, spatial distribution of their synapses and strength of their excitatory drive to motoneurons. V2a interneurons belonging to the first type fire in bursts of action potentials and exhibit unidirectional descending axonal projections that target the distal dendritic region of motoneurons, producing strong excitatory drive. Those of the second type fire with regular action potentials, display bidirectional axonal projections that make synaptic contact at the somatic region of motoneurons, and elicit

weak excitatory drive. The two V2a interneuron types are differentially distributed across the three CPG speed modules. The slow module is mainly driven by excitation from the bursting-type V2a interneurons, whereas the fast module relies primarily on excitatory inputs from non-bursting-type interneurons. The intermediate module displays a hybrid construction and is driven equally from bursting- and non-bursting-type V2a interneurons. This indicates that the three modules do not obey the same circuit configuration rules, but rather their assembly is tailored in a module-specific manner, enabling them to process synaptic information according to the speed and force of the movements they generate (Fig. 7). Thus, our results reveal how the diversity of V2a interneurons determines the logic of the locomotor CPG construction in a task-dependent manner.

Studies in mouse have also shown that V2a interneurons can have heterogeneous firing properties[38–40]. In addition, some V2a interneurons display rhythmic activity during drug-induced locomotion, whilst others do not receive any locomotor-related synaptic inputs[41]. Anatomical evidence shows that V2a interneurons make close contacts with commissural interneurons and motoneurons[42]. However, there is still no information on the exact pattern of activity of V2a interneurons, and their connectivity with motoneurons innervating slow vs fast muscle fibers in adult mouse. It is plausible that the functional diversity of V2a interneurons and their preferential connectivity with slow vs fast motoneurons revealed in our study could also apply in the mammalian locomotor CPG. This can only be directly determined using dual intracellular recordings from functionally identified V2a interneurons and motoneurons. This has not yet been possible in adult mouse spinal cord.

Our analysis reveals the interplay between the diversity of V2a interneurons, their assembly into modular circuits, and their cellular and synaptic properties, which are synergistically tuned in a module-specific manner according to the vigor of the locomotor movement they control. The module encoding slow, less vigorous locomotor movements relies on a circuit design in which convergence of a small number of bursting-type V2a interneurons is sufficient to recruit slow motoneurons. This module is supplemented by a strong potentiation in their respective synaptic connections, reliant on a non-linear integration mode whereby repetitive high frequency firing during V2a interneuron bursts induces powerful dendritic non-linearity in the excitation provided to motoneurons, which is mediated by activation of NMDA receptors. This endows the slow module with a mechanism of amplification of the excitatory drive, resulting in short-term potentiation that enhances the excitation and consequently scales down the number of converging V2a interneurons necessary to recruit slow motoneurons. In contrast, the module encoding fast, more vigorous locomotor movements relies on a large number of converging non-bursting-type V2a interneurons and fast motoneurons, with less involvement of NMDA-dependent non-linearity. The fast module relies primarily on a linear integration mode with a narrow time window for coincidence detection. Therefore, synchronized inputs from a large number of converging non-bursting is required to recruit fast motoneurons, reflecting the behavioral requirement of the fast module for rapidly responding to highly salient stimuli that would generate such coincident inputs occurring during escape behavior (e.g., coincident incoming sensory information indicating a predatory strike). The intermediate sub-circuit module relies on excitatory drive from both bursting-type and non-bursting-type V2a interneurons, forming a hybrid integration mode with both non-linear and linear summation. Thus, the diversity of the firing and integrative properties of V2a interneurons are selectively captured and transformed into two different spatial patterns of excitatory drive. This enables the three CPG modules to perform different computations in a task-dependent manner.

Motoneurons are enriched with mechanisms that contribute to non-linear properties and hence control their recruitment. Many studies across vertebrate species have revealed how specific ionic conductances and their modulation can influence the integration of synaptic and firing properties in motoneurons[7,21,22,43–48]. In addition, dendritic non-linearity mediated by persistent inward current modules have been suggested to amplify synaptic inputs to motoneurons. Also, motoneurons make direct synaptic connections with each other to amplify their excitability[49] and can also act retrogradely to influence the activity of the locomotor CPG[50]. Our study shows how the non-linearity induced by dendritic NMDA receptor activation amplifies the motoneuron excitability in a circuit-specific manner, to allow the primacy of recruitment of circuits underlying locomotion at slow speeds. NMDA has been used to elicit locomotor-like activity in many preparations, and one consistent characteristic of the NMDA-induced locomotor rhythm is its slow frequency and the lack of activity at the higher range of frequencies[12,51–58]. The prevalence of NMDA-dependent non-linearly in the slow circuit module compared to the fast one could be the mechanism that restricts the frequency of the NMDA-induced locomotor rhythm.

A key finding of this study is the direct link between the diversity of V2a interneurons, their circuit connectivity with motoneurons and the control of a behavioral outcome. Major efforts are being deployed to map the connectivity of brain circuits[59]. It has been impossible to build up a dynamic framework of the chain of command leading to a behavioral outcome. Our results not only map the connectivity the locomotor CPG, but also define specific circuit and synaptic rules that govern the control of locomotor movement and its vigor.

Analysis of the spinal locomotor CPG has provided many important insights into the mechanisms of development, organization and function of circuits underlying behavior[5,8,11–14,25,26,60–62]. In the mouse spinal cord, broad classes of neurons have been identified based on the transcription factor they express during development, and in a few instances their anatomical connectivity with motoneurons have been examined using trans-synaptic virus tracing[63–66]. Furthermore, the function of some of these neuronal classes have been probed using ablation during early development. In addition, while the diversity of these neuronal classes is beginning to emerge[67–69], it is not known how this diversity accounts for circuit dynamics and ultimately the generation of locomotor movements. While virus-based reconstruction of synaptic connectivity and calcium imaging inform us about the existence of synaptic interactions and activity of neurons, they do not allow for the fine-grained analysis to directly link neuronal diversity to functional circuit assembly at the single neuron level. Therefore, an important insight from our study is the absolute requirement to directly determine the properties of excitatory synapses for understanding the computation of motor circuits. While the emphasis thus far has been on molecular/genetic markers for neuronal subtypes combined with anatomical substrate for their synaptic connections, these analyses alone only provide a static view of the circuit map. Our study thus indicates that an understanding of the functional diversity of neurons and its relevance within the circuit requires a direct assessment of their firing properties, anatomical features, synaptic weights and their mode of integration, ideally in a circuit with measurable output such as those underlying motor actions.

In conclusion, our study reveals a logic of how the diversity of interneuron physiological properties, anatomical features and synaptic integration tailors the construction of the locomotor CPG in the adult zebrafish. The module-specific circuit design enables the locomotor CPG to adjust the excitatory drive to the vigor of the intended movements. The strong similarities and conserved molecular code for neuronal diversity and circuit assembly between fish and mammals suggest that this parsimonious circuit

design, with its embedded non-linear synaptic integration, could also be applicable to the mammalian spinal cord in the control of the vigor of mammalian locomotor movements. Thus, the logic of circuit assembly revealed in our study could represent a general principle for networks in the central nervous system.

## Methods

**Experimental animals**. Zebrafish (*Danio rerio*) were housed in a core facility at the Karolinska Institute according to established procedures. In this study, a zebrafish line, Tg(*Chx10*:GFP), was used. In this line, green fluorescent protein (GFP) was expressed in V2a interneurons and driven by the promotor *Chx10*, hence facilitating the specific targeting of this interneuron population (Supplementary Fig. 1). All experimental protocols were approved by the Animal Research Ethical Committee in Stockholm and were performed following EU guidelines.

**Tracing, morphology and immunohistochemistry**. Adult zebrafish (Tg(*Chx10*: GFP); 8–10 weeks old) of either sex were used to retrogradely label motoneurons. For this zebrafish were anesthetized in 0.03% tricaine methane sulfonate (MS-222, Sigma-Aldrich) and the fluorescent dye Rhodamine-dextran (3000 MW, Molecular Probes) was injected into all muscle types of one body hemi-segment using pins soaked with the dye. The number of V2a interneurons dye-coupled to the slow, intermediate or fast motoneurons, was estimated using the low molecular mass (287 Da) tracer neurobiotin (25%, Vector Laboratories). The tracer was precisely injected under a dissection microscope into a specific muscle type in one body hemi-segment (8th to 10th segment), rostral to the dorsal fin, that corresponds to the segment in which intracellular recordings are done. Slow and fast muscle was selectively injected with the tracer, while the intermediate muscle could not be targeted separately and therefore it was injected together with the slow muscle. The injected animals were allowed to recover overnight to help the transport of the retrograde tracer. They were then deeply anesthetized with 0.03% MS-222 and the spinal cords were dissected out and fixed overnight at 4 °C with 4% paraformaldehyde (PFA) in phosphate buffer (PB, 0.1 M; pH = 7.4). The fixed spinal cords were washed 3–5 times for 10 min in phosphate buffered saline (0.01 M PBS; pH = 7.4). A solution containing 0.1–2% normal horse, donkey or goat serum with 3–5% bovine serum albumin (BSA) and 1% Triton X-100 in PBS was used for 60 min at room temperature to block non-specific protein binding sites. After this, the spinal cords were then incubated with rabbit (1:500; A11122, Life Technologies) or chicken (1:500; ab13970; Abcam) anti-GFP. For double-labeling of GFP and neurobiotin or *Chx10*, the anti-GFP was used together with either streptavidin conjugated to Alexa Fluor 555 (1:500; Invitrogen) or anti-*Chx10*[70] in 1% Triton X-100 in PBS at 4 °C for 72 h. The tissue was subsequently thoroughly rinsed with PBS and incubated with anti-rabbit or anti-chicken Alexa Fluor 488 antibody (1:500; Invitrogen) overnight at 4 °C. After this step, the spinal cords were rinsed five times for 10 min in PBS and mounted in antifade fluorescent mounting medium (Vectashield Hard Set, Vector Labs). Imaging of whole-mount spinal cords was performed using a laser scanning confocal microscope (LSM 800, Zeiss).

**Electrophysiology**. The ex-vivo preparation of the adult zebrafish (Tg(*Chx10*: GFP), 8–10 weeks old) was used for electrophysiological recordings from synaptically connected V2a interneurons and motoneurons. Fish were anesthetized in 0.03% tricaine methane sulfonate (MS-222, Sigma-Aldrich). Retrograde labeling of spinal motoneurons was achieved by injecting dextran cascade blue (3000 MW, Molecular Probes) into the different muscle types. These animals were kept overnight to allow the transport of the tracer. They were then cold anesthetized in a slush of frozen extracellular solution containing MS-222. The axial musculature was carefully removed up to the caudal end of the dorsal fin, leaving the musculature at the tail intact. The bones were then removed along four to five segments rostral to the dorsal fin to allow access to the spinal cord for recording electrodes. In addition, the vertebral arches of the first two segments of the spinal cord were removed to enable the placement of an extracellular stimulation electrode used to elicit locomotor activity. The ex-vivo brainstem-spinal cord preparation was then placed in a recording chamber that was continuously perfused with an oxygenated extracellular solution containing (in mM): 134 NaCl, 2.9 KCl, 2.1 CaCl$_2$, 1.2 MgCl$_2$, 10 HEPES and 10 glucose with pH 7.8 adjusted with NaOH and an osmolarity of 290 mOsm. The solution was kept at room temperature of 20–22 °C and muscle contractions was blocked using d-tubocurarine (10 μM, Sigma-Aldrich) for the whole duration of the experiment.

Locomotor activity was induced by extracellular stimulation (10 pulses at 1 or 2 Hz) using a suction electrode that was placed at the junction between the brainstem and spinal cord. In some experiments, the extracellular stimulation was delivered close to the Mauthner cell region to produce fast swimming during which fast motoneurons become recruited. Dextran-labeled motoneurons and GFP-positive V2a interneurons were visualized using a fluorescence microscope (Axioskop FS Plus; Zeiss) equipped with IR-differential interference contrast (DIC) optics and a CCD camera with frame grabber (Hamamatsu) and could then be targeted specifically. Intracellular recording electrodes were pulled from borosilicate glass (Hilgenberg) by a micropipette puller (P-1000, Sutter Instrument) and filled with intracellular solution contained (in mM): 120 K-gluconate, 5 KCl, 10 HEPES, 4

Mg$_2$ATP, 0.3 Na$_4$GTP, 10 Na-phosphocreatine with pH 7.4 adjusted with KOH and an osmolality of 275 mOsm. Paired whole-cell patch-clamp recordings were performed simultaneously from identified motoneurons (pre-labeled with a retrograde dye) and GFP-positive V2a interneurons. Two patch-clamp electrodes were placed from opposite directions of the preparation and were driven through the meninges into the spinal cord using motorized micromanipulators (Luigs & Neurmann) while applying constant positive pressure. The intracellular signals were recorded in current-clamp with no bias current, they were amplified with a MultiClamp 700B intracellular amplifier (Molecular Devices) and low-pass filtered at 10 kHz. Paired recordings were randomly carried out from motoneurons and V2a interneurons within the same segment of the spinal cord. Only neurons that had stable resting membrane potentials at or below −50 mV, fired action potentials to suprathreshold current injection and showed minimal changes in series resistance (<5%) were included in this study. The modular identity (slow, intermediate or fast) of the recorded V2a interneurons and motoneurons was determined based on their lowest frequency of the locomotor rhythm at which they became recruited[24,31,33,34]. Neurons of the slow module were recruited at all frequencies throughout the locomotor episode. Those of the intermediate became recruited only at locomotor frequencies above 3.5–5 Hz, while neurons of the fast module were only recruited during fast locomotor cycle (>8–10 Hz).

The existence of synaptic connections was determined by eliciting single or bursts of action potentials in the V2a interneurons while monosynaptic postsynaptic potentials (EPSPs) were recorded in the target motoneurons. The illustrated EPSPs from single action potentials are averages of 50 consecutive sweeps and their amplitude was measured as the difference between baseline and EPSP peak. To mimics the burst firing in V2a interneurons, current pulses with increasing duration (1, 3, 5, 7 and 9 ms) were injected in the recorded V2a interneurons to induce burst with 1–5 action potentials. The firing threshold of motoneurons corresponds to the membrane potential as which the d$V$/d$t$ exceeds 10 V/s during an action potential. The input resistance of motoneurons was calculated as the slope of the linear part of the $I$–$V$ curve obtained by application of hyperpolarizing current pulses.

In some experiments, the N-methyl-D-aspartate (NMDA) receptors antagonist (2$R$)-amino-5-phosphonovaleric acid; (2$R$)-amino-5-phosphonopentanoate[71,72] (AP-5; 100 μM; Tocris) was used to block the NMDA component of the EPSPs. In other experiments, NMDA receptors were selectively blocked in the recorded motoneurons by intracellular dialysis of the use-dependent blocker (5 $s$,10 $R$)-(+)-5-Methyl-10,11-dihydro-5$H$-dibenzo[$a,d$]cyclohepten-5,10-imine maleate[73–75] (MK-801; Tocris). This antagonist was added to the intracellular solution (final concentration of 30 μM) to selectively block NMDA receptors only in the recorded motoneuron without affecting the neighboring cells. To ensure that MK-801 only affected the targeted motoneuron, a second motoneuron was simultaneously recorded in the same segment using control intracellular solution. All the electrophysiological data were analyzed by using Spike2 software (Cambridge Electronic Design) or Clampfit (Molecular Devices). The lowest recruitment frequency of motoneurons and V2a interneurons was defined as the slowest burst frequency in a swimming episode at which the neurons discharged action potentials. The peak-to-trough amplitude of the locomotor-related membrane potential oscillations in motoneurons was analyzed just after breaking into neurons (control) and after allowing MK-801 to diffuse into the recorded neuron. For this the voltage traces were smoothed to remove action potentials and retain only the underlying membrane potential oscillations. The amplitude of the oscillations in control and in MK-801 was plotted as a function of the instantaneous frequency or as cumulative distribution.

**Morphological reconstructions and localization of synaptic sites**. The morphology and sites of synaptic connections between V2a interneurons and motoneurons were examined after reconstruction of neurons. Pairs of recorded V2a interneurons and motoneurons were passively filled with 0.25% neurobiotin for post-hoc analysis of their morphology. Spinal cords with neurobiotin-filled neurons were fixed in 4% PFA solution overnight at 4 °C. They were then washed extensively with PBS and incubated in streptavidin conjugated to Alexa Fluor 555 (1:500; Invitrogen) and mouse anti-synaptic vesicle protein 2[76–79] (SV2; 1:500; Developmental Studies Hybridoma Bank) to detect the synaptic contacts between V2a interneurons and motoneurons. The spinal cords were thoroughly rinsed with PBS and incubated with anti-mouse Alexa Fluor 488 antibody (1:500; Invitrogen) overnight at 4 °C before. They were subsequently rinsed in PBS and mounted in antifade fluorescent mounting medium. The neuronal processes (axons and dendrites) of single-labeled neurons were acquired with Z-stacks by a laser scanning confocal microscope (LSM 800, Zeiss), then traced and reconstructed using manual drawing in Adobe Illustrator. The occurrence of synaptic close appositions between axons of V2a interneurons and dendrites or soma of motoneurons, and their colocalization with the synaptic protein SV2 were determined from single optical confocal sections (0.2 μm). The analysis of the morphology of V2a interneurons and the sites of synaptic connections was performed blind regarding the identity of V2a interneurons (bursting-type vs non-bursting-type) and only close apposition sites that colocalized with SV2 were counted as putative synaptic connections. These synaptic sites were determined from single optical sections acquired with laser scanning confocal microscope. The sections were filtered and pseudo-colored for optimal visualization of synaptic protein SV2. In addition, the full morphology of V2a interneurons and motoneurons was reconstructed to determine their full dendritic tree and axonal projections.

**Analysis and statistics**. The statistical details are described in figure legends. The sample size was determined based on numbers ($n \geq 6$ from different animals) reported in previous studies, the animals were not randomized, and no samples were excluded. All statistical analysis was performed in GraphPad Prism (USA) and significance was determined with two-tailed Student's $t$-test, two-way repeated-measures ANOVA, or Kolmogorov-Smirnov test. The homogeneity of variances was tested using F-test and were found similar between sample groups. All values are expressed as means and the error bars reflect s.e.m. and the $n$ number given correspond to different experiments from separate animals.

**Data availability**. The data that support the findings of this study are available from the corresponding authors upon reasonable request.

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

## Acknowledgements

We thank Drs. Sten Grillner, Andreas Kardamakis, Laurence Picton, and Gilad Silberberg, as well as the members of our lab for comments on an early version of the manuscript. We are grateful to Shin-ichi Higashijima for providing the zebrafish line and Chx10 antibody used in this study, and Andreas Kardamakis for help with the analysis shown in Fig. 7. This study was supported by grants from the Swedish Research Council (2017–02905), the Swedish Brain Foundation (FO2018-0306) and the Karolinska Institute (to A.E.M.). J.S. research is supported by the National Natural Science Foundation of China (31771168), the Fundamental Research Funds for the Central Universities (1500219132), the National Key Research and Development Program of China (2016YFA0100800) and the State Key Program of National Natural Science Foundation of China (81330030).

## Author contributions

J.S. designed and carried out experiments, acquired and analyzed the data, and was involved in the writing the manuscript. E.D. performed and analyzed the anatomical

experiments. A.E.M. initiated the project, designed the experiments, analyzed the data and wrote the manuscript.

## Additional information

**Competing interests:** The authors declare no competing interests.

