## [Peer Review File · Nature Communications]

Reviewers' comments:

Reviewer #1 (Remarks to the Author):

This interesting study by Song and ElManira addresses a highly time issue in the neurosciences: which mechanisms in neural networks within the nervous system of animals contribute to motor flexibility. The authors study this question on the adult zebrafish, which offers the chance to profit from the available neurogenetic toolset for identifying premotor interneurons in the spinal cord. They focus in their analysis on the level of information transfer between spinal premotor INs and swim motoneurons in generating different speeds and intensity of swimming activity. In an elegant set of experiments the study shows that an up-to-now unknown diversity of one important population of spinal CPG interneurons for swimming, the V2a-interneurons (V2a-INs), contributes to motor flexibility, i.e. to the sequential recruitment of slow, intermediate and fast motoneurons with increasing swimming intensity. The diversity of V2a-INs is fourfold: firstly, there exist bursting and nonbursting V2a-INs, which show specific connectivity to the three different classes of motoneurons, with bursting V2a-INs connecting more often to slow motoneurons as compared to fast motoneurons. Secondly, the two classes of V2a-INs show different connectivity patterns with motoneurons with respect to the location of their synaptic contacts. Bursting V2a-INs connect relatively more often to the dendrites as compared to the non-bursting, which more often connect to cell bodies. Thirdly, distribution of synapses from V2a-INs onto motoneurons differs between motoneuron classes, with fast motoneurons receiving relatively more dendritic inputs from V2a-INs compared to intermediate and slow motoneurons. Fourthly, V2a-IN to motoneuron synapses express dependent on the motoneuron type they connect to specific short term plasticity. While V2a-INs connecting to slow motoneurons show substantial potentiation of their EPSPs, intermediate and fast motoneurons show only weak or mere nonlinear summation, respectively. The authors provide evidence for this potentiation being dependent on NMDA-receptor action. In a final set of experiments the authors show that the NMDA-dependent non-linearity indeed underlies recruitment of motoneuron types. Summarizing their data collected the authors estimate the number of V2a-INs contributing to the synaptic drive provided to the three different types of motoneurons.

Experiments, structural analysis, data evaluation and statistics have been conducted thoroughly. The presentation of results is generally clear and concise. The interpretation of results is appropriate. This excellent study will be highly interesting for all researchers in the field of motor control.

I have the following specific suggestions that shall either improve clarity of presentation or point to some syntactic or orthographic errors.

1. P7, para 1, li 4-8 and Fig.2: the authors state that EPSP-amplitudes of bursting V2a-INs were largest in the slow and smallest in fast motoneurons. Statistics for this comparison is not presented. Please supply.
2. P7, para 2, li3-7 and Fig.3: information on the number of V2a-INs, which were stained and evaluated morphologically for the different types of V2a-INs is not given. Please provide this information.
3. P7, para 2, li 9-12: "Slow motor neurons preferentially received dendritic inputs from bursting V2a-interneurons (82% of connections), whereas fast motor neurons received synaptic inputs mostly on their somatic region from non-bursting V2a-interneurons (25% of the connections)." This formulation is a bit misleading: the reader may take away that only 25% of the synapses are found on the somatic region. I suggest to say "(only 25% of the connections on the dendrites)".
4. P8, para 2, li 11: "recruitment frequency" – do you mean "swimming frequency" or "action potential frequency"?
5. P9, last li: "motor neurons" instead of "moto neurons" and "potentiation" instead of "potential"
6. P10, para 2, li 13: "(Fig.6a,f)" - shouldn't it be "(Fig.6a-f)"?
7. P11, para 2, li 7: "needed" instead of "need"
8. P11, para 2, li 8-11: when calculating the number of V2a-INs necessary to bring motoneurons up to

their firing threshold, isn't it also necessary to include the number of action potentials generated by the INs? I could not find information regarding this in the text.

9. P11, para 2, line 2 from bottom: Shouldn't it read "For the non-linear....." instead of "the non-linear"?

10. Figure 6: in part c and f the cumulative distribution of amplitudes of depolarization is plotted. The max value for both conditions should reach 1.0 for the max amplitude value. What is the reason for the horizontal line of filled circles (c in red, f in green) at the level of 1.0? Does the distribution in the control condition show bifurcation? Then, however the max value should not be 1.0, but less.

11. Figure 7: this figure is very important as it shall summarize the data collected conceptually. It is probably an elegant approach the authors choose for estimating the number of V2a-INs needed to drive each of the three classes of motoneurons. Even though I read the text several times and tried to dissect the figure, I was not able to completely follow the reasoning of the authors in subfigures a and b. To my opinion more information is needed for the reader to follow the line of thoughts of the authors. I suggest that the authors generate a figure, in which they show at least for one of the motoneuron types, how data points in a and b were generated.

12. P15, li 10-20: from the discussion it reads that recruitment of motoneurons from slow to intermediate and fast determines intensity of locomotion and thereby also speed. What does this mean for an interplay between motoneurons being recruited supra-thresholdly with cycle period of locomotor activity and swimming speed?

13. P21, para 2, li 9-11: maybe I missed it, but I did not find reference to the study showing that MK-801 blocks NMDA-receptors selectively. The same is true for the general NMDA-receptor antagonist AP-5.

Reviewer #2 (Remarks to the Author):

Song and El Manira's manuscript provide a very interesting dataset showing diversity in a speed module dependent manner in short term plasticity at the synapses between excitatory premotor V2a interneurons and motor neurons in zebrafish juvenile/adult spinal cord. Such circuit organization in the spinal cord could be highly relevant to set the vigor of locomotor movements.

The data provided in the manuscript is of great interest for building in the near future comprehensive models of recruitment of motor neurons according to speed. Yet, multiple points need be specified and strengthened:

A thorough characterization of the Tg(ax:GFP) line is necessary at the juvenile/adult stage used here. Second, voltage clamp recordings of MNs are critical to establish the mechanisms underlying the short-term plasticity observed and its NMDA dependence. Furthermore, there is a contraction that the authors need to explain between Figure 1 and 7 of the manuscript. While the authors initially describe more presynaptic V2a partners onto slow (S, ~3.5) MNs than intermediary (I, ~2.5) or fast (F, ~2.5) MNs, their estimates of the number of V2a necessary to spike MNs for each category are following the opposite trend: between 4 (for non bursting) and 9 times (for bursting V2a & NMDA dependent STP) more V2a INs are necessary to spike F MNs compared to S MNs. How do the authors explain this contraction? Isn't it the demonstration that dye coupling is biased for small cells such as S MNs and work poorly for detecting inputs converging onto large MNs? An alternative explanation would be that electrical coupling necessary for dye coupling is biased for S V2a- S MNs pairs. Finally, the data provided in Figure 3 showing overlap between SV2 labeling & neurobiotin is not convincing as it is.

One general comment for introduction and discussion: overall the authors only focus on convergence of V2a inputs for triggering spiking of motor neurons, but they do not refer to the concept of divergence that one V2a can project onto multiple MN targets-possibly in some cases from different speed-specific modules (? see comment below).

Major comments

Figure 1. Distribution of motor neurons and V2a interneurons.

a) While motor neurons were labeled using retrograde tracers injected in all muscles, the critical quantification of V2a interneurons per segment rely on counting neurons in the Tg(alx:GFP) line generated by Kimura et al 2006. Generated using BAC transgenesis prior to the time of using Crispr-Cas mediated and targeted genome-editing, this transgenic line has at the larval stage non-specific expression in non V2a neurons and may not label all V2a interneurons.

In particular, is there any overlap between motor neurons and the Tg(alx:GFP) at this stage? The conclusions on convergence from V2a onto MNs rely on the specificity and coverage of GFP to the entire V2a population by the transgene Tg(alx:GFP).

As a quantitative assessment is missing at the juvenile/adult stage, the authors should demonstrate that the Tg(alx:GFP) line is specific for V2a interneurons by combining IHC or FISH on C hx10 together with IHC against GFP. The authors should add in Panel 1b the degree of overlap between the MN labeling and GFP expression to verify that no MN express GFP in the transgenic line used.

b) One related question: when targeting slow or fast muscles for retrograde labeling, did the authors see when monitoring dye coupling from motor neurons that other non V2a cells were labeled? If there was some diversity, it would be useful to summarize in a table all the information on cells labeled (number, type, location etc) that are included in these dye coupling experiments.

c) As the same authors mentioned in a previous study the existence of electrical coupling among V2as as well as between MNs (Song et al Nature 2016, one might expect that many motor neurons should pick up the dye – with possibly a larger population of V2as than just the presynaptic ones to the motor neurons labeled. Please comment.

d) As described in the methods, intermediary muscle fibers (I) could only be labeled together with slow ones (S). If ones attempt to subtract values from S versus S+I, it seems that intermediary MNs would receive from only 2.5 V2a INs – similarly to fast MNs and under S MNS (3.5). 2 points:
1- Are S MNs receiving inputs from more V2a INs or is the dye coupling mechanisms possibly biased for S MNs and therefore not reflecting the amount of presynaptic neurons coupled by electrical synapses?
2- How does this observation (panels 1e and 1h) can be reconciled with the fact that degree of convergence needs to be much higher for I and F V2a INs onto MNs than for S V2a INs onto MNs based on subsequent estimations performed by the authors (and intuitive deduction that less resistive cells will need more inputs to be recruited if EPSCs are the same – see first comment of Figure 2)?

e) Panel a & b on how many segments is this analysis based and in where in the rostral caudal dimension of the spinal cord? The data shown here looks similar to the previous report of Menelaou et al 2013 - this publication should be therefore referenced as it describes the same V2a morphological features.

Figure 2. Functional diversity of V2a interneurons and their synaptic drive to motor neurons.

a) Are the differences in EPSP amplitude elicited after V2a single spike solely due to difference in input resistance? The authors should perform voltage clamp recordings of MNs to demonstrate whether

EPSCs are similar or not in amplitude in slow, intermediary and fast MNs. This is a critical information to build a model.

b) Amplitudes of example EPSP for bursting V2a in panels 2b and 2c are not reflecting the average shown in the quantification.

c) Schematic in panel 2e seems to indicate that the connectivity is strict from F V2a to F MN, I V2a to I MN and S V2a to S MN. Isn't it misleading? Please provide the percentage of pairs matching this scheme.

Figure 3. Distinct morphological features of the two types of V2a interneurons.

The methods need to be specified more clearly for this figure and data should be more convincing. As it is, the data does not clearly support the model drawn in panel 2c.

a) Tracing was performed apparently on projections from Z stacks in AI. Can the authors confirm that putative site of connections were assessed in the Z stacks and were performed blind regarding the type of V2a and MN recorded?

b) The overlap between neurobiotin and SV2 on soma or axon is not clear on the saturated images shown in panels 3a and 3b. Is the SV2 labeling specific? The authors should show non-saturated images prior to adjustment of contrast and brightness. Overlap should be quantified in an unbiased manner.

Figure 4. NMDA-dependent short-term plasticity of bursting-type V2a interneuron synaptic transmission.

a) Recording in V-clamp of MNs are critical to demonstrate that post synaptic NMDA receptors are responsible for the effect of STP observed here. If the cell is clamped at -80mV , there should not be potentiation of the bursting V2a-MN synapses.

b) Need to provide some statistical analysis to further support the effect of NMDA channel block.

c) Also given the fact that AP5 had only a modest effect (especially on the fast module) it could be interpreted that F MN actually receive a biased electrical synaptic info from F V2a. This is not addressed in the manuscript.

Figure 5. Absence of NMDA-dependent short-term plasticity in non-bursting-type V2a interneurons.

a) Same comment as in Fig 4 - looks like there is no effect of AP5 (please provide statistical analysis for panels c&d). This would suggest that non-bursting V2a form predominantly electrical connections?

Figure 6. NMDA-dependent non-linearity determines the recruitment of motor neurons.

a) How do the authors explain the reduction in firing of slow MNs with blockers of NMDA receptors. Could it be that slow MNs form NMDA-dependent autapses that contribute to the spiking? Are slow MNs spiking classical sodium action potentials? It is not obvious from these panels.

b) Again it is critical to show V Clamp recording of MNs at hyperpolarized potential where NMDA receptors should not be recruited

c) Same comment as above: please provide statistical analysis for panels in this figure. Also please provide numbers of cells recorded and animals used for each figure.

d) Panel g: why is this fast MN virtually silent during your stimulation paradigm? The authors must have elicited fast swimming to identify its recruitment at high locomotor frequency, why isn't it shown here for both conditions of control and MK801? In order to conclude that the "recruitment of fast motor neurons during swimming does not require activation of NMDA activity", the authors need to show recruitment with spiking of F MNs in control conditions.

Figure 7. Module-specific convergence pattern of bursting- and non-bursting-type V2a interneurons. The manner this figure has been designed seems misleading for the reader: these estimates are not based on extensive paired recordings that would constitute a full demonstration. Therefore it should be explicit in the text and axis of bar graphs that the values provided are estimations only.

Furthermore, a detailed procedure of how spiking threshold and integration of the number of inputs depending on their location on the soma or dendrites to trigger spiking should be added to the methods. Rough estimates based on summations of EPSP amplitudes at resting potential are providing a useful basis but they neglect the biophysical properties of motor neurons themselves regarding summation of EPSPs, which is by itself a complex problem requiring a fine analysis of the dynamics of EPSCs and biophysical properties of the postsynaptic targets.

Minor comments

- Figure 1 panel: indicate how V2a or MNs were labeled on the panel or at least in the legend- information is lacking to fully appreciate the data.
- Figure 1. Panel a V2a label - is this a Z stack projection of a whole segment? if yes why are there only 9 cells visible, which is not matching the quantification given by the authors. Since the analysis is done per hemi segment it is critical to show a representative projection of the entire hemisegment for this region where the segment borders are highlighted and all labeled cells are shown.
- Legend Figure 1. Missing a word "Number of intersegmental V2a interneurons the 4 adjacent segments (2 rostral and 2 caudal) converging onto slow, intermediate and fast motor neurons"
- Figure 2. Panel D - how many EPSPs were analysed for each speed module? Would be easier to see individual values plotted as a scatter rather than a bar chart. Also what is the failure rate and did the authors average traces in case of failure or not? Does every V2a AP result in an EPSP in the paired MN?
- Figure 3. Here the authors only show the relative % of synapses targeting the dendritic vs the somatic compartments, however the data can be further unpacked. For example it would be interesting to see what are the actual total numbers of V2a synapses on a given slow vs fast MN? Are there any differences? Also you could check the average no of synapses made by a bursting vs a non bursting V2a?

Reviewer #3 (Remarks to the Author):

In this study, the authors use the adult zebrafish preparation in order to study the relationship of a class of spinal interneurons (INs), V2a INs, and motoneurons (MNs). This group has previously demonstrated that V2a INs are "matched" to MNs, in that there are 3 classes of INs each innervating MNs that in turn innervate slow, intermediate, and fast muscle fibres. In this manuscript, the authors now demonstrate that within these classes, V2a INs can be further classified into bursting vs non-bursting INs, with the bursting INs innervating MN dendrites where NMDA receptors lead to non-linear EPSP summation, and non-bursting V2a INs that innervate MN somata to produce linearly summing EPSPs. Furthermore, the proportion of bursting vs non-bursting V2a INs innervating MNs systematically changes from slow to medium to fast MNs, thus leading to a logic that could produce appropriate behavioural responses.

This paper demonstrates the complexity of neural circuits for movement – even at this most basic

(spinal cord!, zebrafish!) level. To those in the field, this complexity may not be surprising, but its demonstration will be welcome and will undoubtedly lead to new experimental approaches. To those not in the spinal cord physiology field, I think this manuscript will be enlightening and demonstrate that understanding the connectome alone without understanding neuronal and synaptic physiology will lead to an alarmingly incomplete understanding of the brain. While this view is well appreciated to invertebrate neurobiologists (and was always eloquently stated by Peter Getting in the 70s and 80s), this clear demonstration in a vertebrate model should provoke some introspection. So I think this work will have widespread appeal.

However, I do have some concerns, particularly about the clarity in parts of the manuscript. I think that addressing these would make this an even stronger paper.

1. Results related to convergence, Figure 1: Here, the authors have used retrograde labeling of MNs to count the number of MNs in a segment and relate this to the number of V2a INs either in that hemisegment or adjacent hemisegments. As V2a INs form electrotonic synapses with MNs, the dye is transported and double labelled V2a INs can be counted. They then demonstrate that there are fewer V2a INs than the number of MNs they innervate. If there is convergence of V2a INs onto MNs, then we should see more than one V2a IN labelled when a single MN is filled, for example. If every MN receives V2a input, then the data would support that there is divergence of V2a INs – that a single V2a IN innervates more than one MN. But do they support convergence – that more than one V2a IN projects to a single MN? What do the numbers in Fig 1e and Fig 1h mean – that is, how were they derived? (On a minor related note, do 100% of V2a INs form electrotonic synapses with MNs? And are these both bursting and non-bursting types?) Convergence is a key point (see Fig 7), so should be made crystal clear.

2. Plasticity. The authors use this term repeatedly, and it is unclear what they mean. If they are referring to non-linear summation of EPSPs mediated by NMDA receptors, I would not call it plasticity. As it stands, I think the term is a distraction at best, and does not add to the important findings of this manuscript. At minimum, their use of the term should be clearly defined and demonstrated.

Minor comments:

1. Last sentence on p.4 has no predicate.

2. p.6, top paragraph, last sentence ("In addition..."): the relevance/meaning of this sentence is opaque.

3. p.7, last few lines: 82% of slow MNs have dendritic inputs, so this is most, whereas fast MNs have 25% somatic – this is not most.

4. I'm not sure that Fig 2E contributes

5. p.10, middle – delay the fast module? Why a time delay here?

6. Fig 6, amplitude graphs: it is not clear how the amplitude was calculated given the neurons are spiking – are the spikes included in the amplitude calculations? Is this reasonable? Is it meaningful?

7. Top paragraph p.12 – it would be helpful if the number of neurons was reported throughout, not just in the first instance.

8. p.12 – 31 V2a INs is still a huge number considering the number in the hemisegment... the authors may want to remind the reader of this here.

9. The authors refer to morphological features of V2a INs throughout – do they mean axonal projections? If other morphological features were quantified, they should be presented. Otherwise, the term should be clarified.

10. Given the dendritic non-linearities in MNs, even though they're shown to be NMDA receptor dependent, perhaps a mention of the extensive literature on dendritic non-linearities in mammalian MNs would be appropriate? Also, Brownstone et al 1994 showed non-linear increase in MN excitation during cat locomotion – this could be relevant?

11. SEM – In my view, there is little indication to use the SEM in biology, in which we're interested in variability. For eg, in Fig 2D, it would be helpful to show scatter plots or box-whisker plots, or at minimum show and report the standard deviation rather than the SEM. But I do feel like I'm blowing against the wind when I raise this point in 9 out of 10 reviews at least!

We are grateful to the Reviewers for their constructive criticisms and helpful comments and suggestions. We have performed additional experiments and revised the paper to take into account their comments and suggestions as detailed below.

Reviewer #1

1. P7, para 1, li 4-8 and Fig.2: the authors state that EPSP-amplitudes of bursting V2a-INs were largest in the slow and smallest in fast motoneurons. Statistics for this comparison is not presented. Please supply.

The statistics for this comparison have been added on page 7, lines 14-16.

2. P7, para 2, li3-7 and Fig.3: information on the number of V2a-INs, which were stained and evaluated morphologically for the different types of V2a-INs is not given. Please provide this information.

The number of V2a INs, which were stained and evaluated morphologically, has been added on page 8, lines 6, 8, 10 and 11.

3. P7, para 2, li 9-12: “Slow motor neurons preferentially received dendritic inputs from bursting V2a-INs (82% of connections), whereas fast motor neurons received synaptic inputs mostly on their somatic region from non-bursting V2a-INs (25% of the connections).” This formulation is a bit misleading: the reader may take away that only 25% of the synapses are found on the somatic region. I suggest to say “(only 25% of the connections on the dendrites)”.

This was corrected as suggested by the Reviewer.

4. P8, para 2, li 11: “recruitment frequency” – do you mean “swimming frequency” or “action potential frequency”?

This refers to recruitment frequency during swimming. This has been clarified on page 9, lines 4-7.

5. P9, last li: “motor neurons” instead of “moto neurons” and “potentiation” instead of “potential”

6. P10, para 2, li 13: “(Fig.6a,f)” - shouldn’t it be “(Fig.6a-f)”?

7. P11, para 2, li 7: “needed” instead of “need”

These mistakes have been corrected – thank you.

8. P11, para 2, li 8-11: when calculating the number of V2a-INs necessary to bring motoneurons up to their firing threshold, isn’t it also necessary to include the number of action potentials generated by the INs? I could not find information regarding this in the text.

Yes, the number of action potentials generated by the V2a INs were included in calculating the number of INs necessary to bring MNs up to their firing threshold. On average V2a INs fire 3 action potentials during each locomotor cycle (Ausborn et al. 2012) and therefore we

divided the number of EPSPs by 3 to estimate the number of converging INs. This information is now included in the Results (page 12, last 4 lines).

9. P11, para 2, line 2 from bottom: Shouldn't it read "For the non-linear....." instead of "the non-linear"?

This was corrected – thank you.

10. Figure 6: in part c and f the cumulative distribution of amplitudes of depolarization is plotted. The max value for both conditions should reach 1.0 for the max amplitude value. What is the reason for the horizontal line of filled circles (c in red, f in green) at the level of 1.0? Does the distribution in the control condition show bifurcation? Then, however the max value should not be 1.0, but less.

We have now reanalyzed the data for the two conditions separately to avoid this issue. Thank you for pointing this out. The graphs in Fig. 6 have been modified accordingly.

11. Figure 7: this figure is very important as it shall summarize the data collected conceptually. It is probably an elegant approach the authors choose for estimating the number of V2a-INs needed to drive each of the three classes of motoneurons. Even though I read the text several times and tried to dissect the figure, I was not able to completely follow the reasoning of the authors in subfigures a and b. To my opinion more information is needed for the reader to follow the line of thoughts of the authors. I suggest that the authors generate a figure, in which they show at least for one of the motoneuron types, how data points in a and b were generated.

We have revised the text to explain better the reasoning and how the convergence of V2a INs onto the different MN pools have been estimated (pages 12-14).

12. P15, li 10-20: from the discussion it reads that recruitment of motoneurons from slow to intermediate and fast determines intensity of locomotion and thereby also speed. What does this mean for an interplay between motoneurons being recruited supra-thresholdly with cycle period of locomotor activity and swimming speed?

We are not entirely sure we understand the question of the Reviewer.

The slow, intermediate and fast MNs are added successively as the speed and amplitude of swimming increase. The already recruited MNs remain active as additional ones are added. There is no de-recruitment of MNs with the increase of swimming speed.

13. P21, para 2, li 9-11: maybe I missed it, but I did not find reference to the study showing that MK-801 blocks NMDA-receptors selectively. The same is true for the general NMDA-receptor antagonist AP-5.

These references have now been added.

Reviewer #2

Major comments

Figure 1. Distribution of motor neurons and V2a INs.

a) While motor neurons were labeled using retrograde tracers injected in all muscles, the critical quantification of V2a INs per segment rely on counting neurons in the Tg(alx:GFP) line generated by Kimura et al 2006. Generated using BAC transgenesis prior to the time of using Crispr-Cas mediated and targeted genome-editing, this transgenic line has at the larval stage non-specific expression in non V2a neurons and may not label all V2a INs.

It is known that Chx10 actively promotes V2a fate, downstream of the LIM-homeodomain factor Lhx3, while concomitantly suppressing the MN developmental program (Clovis et al. Cell Reports, 2016). Conversely, Hb9 is also known to promote MN development by inhibiting V2a IN fate via suppression of Chx10 expression (Lee et al. Dev. Cell, 2008). Knock out of Chx10 or Hb9 results in ectopic induction of MN or V2a IN development program, respectively. It is not surprising that there could be some weak expression of Chx10 during early development in some MN that is fully suppressed once these MNs acquire their mature fate. Indeed, in the adult zebrafish we have never detected any expression of GFP in retrogradely labeled MNs (see page 5, lines 6-9 from bottom)

In particular, is there any overlap between motor neurons and the Tg(alx:GFP) at this stage?

We have done hundreds of retrograde labeling of MNs using dextran-coupled dyes in the Tg(alx:GFP) at the adult stage, we never detected any GFP signal in retrogradely labeled MNs. In this transgenic line, there is no overlap of the expression of GFP in retrogradely labeled MNs.

In addition, we have recorded and reconstructed the morphology of hundreds of INs using the Tg(alx:GFP). All our results show that the recorded INs have morphologies similar to those of the CiD INs, i.e. they are ipsilaterally projecting with a descending axonal projection. These INs always make excitatory connections with MNs.

The conclusions on convergence from V2a onto MNs rely on the specificity and coverage of GFP to the entire V2a population by the transgene Tg(alx:GFP).

All controls done on this transgenic line combined with the extensive recordings and reconstructions of the GFP-labeled INs confirm that the expression of GFP is restricted to Chx10-expressing V2a INs. These INs have descending axons and some also have also an ascending axonal projection. The number of V2a INs per segment reported here in adult zebrafish is similar to that in larval zebrafish (>5dpf), indicating that the number of INs does not change during development. Thus, the transgenic line used in this study is selectively expressing GFP in V2a INs and covers the same number of INs in larval and adult stages.

As a quantitative assessment is missing at the juvenile/adult stage, the authors should demonstrate that the Tg(alx:GFP) line is specific for V2a INs by combining IHC or FISH on Chx10 together with IHC against GFP. The authors should add in Panel 1b the degree

of overlap between the MN labeling and GFP expression to verify that no MN express GFP in the transgenic line used.

We have now performed the IHC control experiment suggested by the Reviewer. The results shown in Supplementary Fig. 1 show that practically all GFP-expressing V2a INs co-localize Chx10. These data are referred to on (page 5, lines 6-9 from bottom).

b) One related question: when targeting slow or fast muscles for retrograde labeling, did the authors see when monitoring dye coupling from motor neurons that other non V2a cells were labeled? If there was some diversity, it would be useful to summarize in a table all the information on cells labeled (number, type, location etc) that are included in these dye coupling experiments.

This is an interesting point. However, we can only identify the V2a INs based on their co-labeling with neurobiotin and GFP. Other potential non-V2a INs cannot be distinguish from retrogradely labeled MNs. Therefore, it is not possible to provide the information requested by the Reviewer.

c) As the same authors mentioned in a previous study the existence of electrical coupling among V2as as well as between MNs (Song et al Nature 2016, one might expect that many motor neurons should pick up the dye – with possibly a larger population of V2as than just the presynaptic ones to the motor neurons labeled. Please comment.

MNs of each speed-module (slow, intermediate and fast) are targeted by the same V2a INs (Ampatzis et al. 2014). Therefore, the potential diffusion of dye electrically-coupled MNs should not result in a large increase in the number of dye-coupled V2a INs.

d) As described in the methods, intermediary muscle fibers (I) could only be labeled together with slow ones (S). If ones attempt to subtract values from S versus S+I, it seems that intermediary MNs would receive from only 2.5 V2a INs – similarly to fast MNs and under S MNS (3.5). 2 points:

1- Are S MNs receiving inputs from more V2a INs or is the dye coupling mechanisms possibly biased for S MNs and therefore not reflecting the amount of presynaptic neurons coupled by electrical synapses?

The number of dye-coupled V2a INs was in the same range in all types of MNs. The number of dye-coupled V2a INs ranged between 2 and 4 from slow MNs, 5 and 7 from slow/intermediate and 2 to 3 from fast MNs.

We have now added data of dye-coupling by injection of neurobiotin in single MNs (new Supplementary Fig. 2). The data show that the range of dye-coupled V2a INs was similar in the three MN types. This argues against any bias in dye-coupling from a given MN type. This is now stated on page 6, paragraph 1, last two sentences.

2- How does this observation (panels 1e and 1h) can be reconciled with the fact that degree of convergence needs to be much higher for I and F V2a INs onto MNs than for S V2a INs onto MNs based on subsequent estimations performed by the authors (and intuitive deduction that less resistive cells will need more inputs to be recruited if EPSCs are the same – see first comment of Figure 2)?

The dye-coupling data provide an anatomical estimation, albeit an underestimation, of the potential number of V2a INs converging on MNs of the slow, intermediate and fast modules. For the slow and intermediate MNs, the estimated number of the converging V2a INs necessary to drive the different MNs to firing threshold (Figure 7) is within the range of the number of converging obtained by dye-coupling.

For fast MNs, the number of V2a INs necessary to drive these MNs to firing threshold largely exceeds the number of converging INs estimated by dye-coupling. This indicates that the recruitment of fast MNs requires highly salient stimuli that would generate coincident excitatory inputs as during escape behavior. This is now clearly indicated in the Discussion (page 17, lines 10-14).

e) Panel a & b on how many segments is this analysis based and in where in the rostral caudal dimension of the spinal cord?

The injections were done in a specific segment rostral to the dorsal fin (8th to 10th segment) that correspond to segments where intracellular recordings were performed. The number of MNs and V2a INs was analyzed in the same segment. This is now indicated in the Methods (page 20, paragraph 2, lines 7-9).

The data shown here looks similar to the previous report of Menelaou et al 2013 - this publication should be therefore referenced as it describes the same V2a morphological features.

We now refer to Menelaou et al. 2014 in this section of the Results.

Figure 2. Functional diversity of V2a INs and their synaptic drive to motor neurons.

a) Are the differences in EPSP amplitude elicited after V2a single spike solely due to difference in input resistance? The authors should perform voltage clamp recordings of MNs to demonstrate whether EPSCs are similar or not in amplitude in slow, intermediary and fast MNs. This is a critical information to build a model.

We have now added data showing that there is no correlation between the EPSP amplitude elicited by V2a single spike and the input resistance of MNs (new Fig. 2e and 2f). This indicates that the different in the EPSP amplitude is not solely due to a difference in input resistance.

b) Amplitudes of example EPSP for bursting V2a in panels 2b and 2c are not reflecting the average shown in the quantification.

We have changed the example EPSPs as requested. We also show 10 superimposed EPSPs for each condition and an average of all the recorded sweeps (50 sweeps). In addition, the quantification is now represented as box and whiskers as requested by Reviewer #3.

c) Schematic in panel 2e seems to indicate that the connectivity is strict from F V2a to F MN, I V2a to I MN and S V2a to S MN. Isn't it misleading? Please provide the percentage of pairs matching this scheme.

The schematic has been omitted as requested by Reviewer #3.

Figure 3. Distinct morphological features of the two types of V2a INs.

The methods need to be specified more clearly for this figure and data should be more convincing. As it is, the data does not clearly support the model drawn in panel 2c.

The methods have been specified in detail now on page 25, line 5-17.

a) Tracing was performed apparently on projections from Z stacks in AI. Can the authors confirm that putative site of connections were assessed in the Z stacks and were performed blind regarding the type of V2a and MN recorded?

Yes, the putative synaptic sites were assessed in the Z stacks and the analysis was performed blindly, i.e. the observer had no information, regarding the type of V2 INs and MNs to be analyzed (page 25, line 5-17).

b) The overlap between neurobiotin and SV2 on soma or axon is not clear on the saturated images shown in panels 3a and 3b. Is the SV2 labeling specific? The authors should show non-saturated images prior to adjustment of contrast and brightness. Overlap should be quantified in an unbiased manner.

The SV2 antibody (from Developmental Studies Hybridoma Bank) was raised to purified synaptic vesicles from the electric fish organ. This antibody specifically labels presynaptic terminals in fish, frog and mammals (Buckeley and Kelly, 1985). It has been extensively used in zebrafish (Reimer et al., 2008; Hao et al., 2012; Stil et al., 2015). These references are now added in the Methods.

The sites of synaptic contact and their co-localization with SV2 was done blind regarding the type of V2a INs and MNs. This is now stated in the Methods.

Figure 4. NMDA-dependent short-term plasticity of bursting-type V2a IN synaptic transmission.

a) Recording in V-clamp of MNs are critical to demonstrate that post synaptic NMDA receptors are responsible for the effect of STP observed here. If the cell is clamped at -80mV, there should not be potentiation of the bursting V2a-MN synapses.

The fact that the non-linear summation was blocked by AP5 indicate that NMDA receptors and that intracellular MK-801 decreased the excitatory drive and recruitment of slow and intermediate MNs indicates that NMDA receptors are post-synaptic. Therefore, we do not feel V-clamp experiments are critical for demonstrating this point.

b) Need to provide some statistical analysis to further support the effect of NMDA channel block.

Statistical analysis is now provided as requested.

c) Also given the fact that AP5 had only a modest effect (especially on the fast module) it could be interpreted that F MN actually receive a biased electrical synaptic info from F V2a. This is not addressed in the manuscript.

The NMDA-induced non-linearity was not affected by the presence or absence of electrical coupling irrespective of the MN type as shown in Supplementary Fig. 3.

Figure 5. Absence of NMDA-dependent short-term plasticity in non-bursting-type V2a INs.

a) Same comment as in Fig 4 - looks like there is no effect of AP5 (please provide statistical analysis for panels c&d). This would suggest that non-bursting V2a form predominantly electrical connections?

Statistical analysis is now provided in Fig. 5.

The proportion of mixed vs chemical synapses was similar in bursting and non-bursting V2a. We have also added a new Supplementary Fig. 3 showing the effect on AP5 on mixed vs chemical synaptic transmission between non-bursting V2a INs and MNs.

Figure 6. NMDA-dependent non-linearity determines the recruitment of motor neurons.

a) How do the authors explain the reduction in firing of slow MNs with blockers of NMDA receptors. Could it be that slow MNs form NMDA-dependent autapses that contribute to the spiking?.

From our extensive recordings of slow MNs, we have no evidence for the existence of autapses.

Are slow MNs spiking classical sodium action potentials? It s not obvious from these panels

Yes, the spikes in slow MN are mediated by voltage-gated sodium channels.

b) Again it is critical to show V Clamp recording of MNs at hyperpolarized potential where NMDA receptors should not be recruited

Selective blockade of NMDA receptors in single MNs by intracellular MK-801 supports the role of NMDA receptors for the recruitment of slow and intermediate MNs. While V-clamp could further support our conclusion, it does not seem critical in these experiments.

c) Same comment as above: please provide statistical analysis for panels in this figure. Also please provide numbers of cells recorded and animals used for each figure.

Statistical analysis is now provided. In these experiments, we used one MN per animal. The number of neurons and animals is now provided in the Figure legend.

d) Panel g: why is this fast MN virtually silent during your stimulation paradigm? The authors must have elicited fast swimming to identify its recruitment at high locomotor frequency, why isn't it shown here for both conditions of control and MK801 ? In order to conclude that the 'recruitment of fast motor neurons during swimming does not require

activation of NMDA activity”, the authors need to show recruitment with spiking of F MNs in control conditions.

We agree with the Reviewer and that is why we have shown two examples of fast MNs. One where the MN is not recruited (Fig. 6g-i) and the other where the MN is recruited (Fig. 6j-l). MK-801 had no effect neither on the subthreshold nor on the supra-threshold activity of fast MNs. This is now clearly stated in the text on page 11, lines 10-14.

Figure 7. Module-specific convergence pattern of bursting- and non-bursting-type V2a INs.

The manner this figure has been designed seems misleading for the reader: these estimates are not based on extensive paired recordings that would constitute a full demonstration. Therefore it should be explicit in the text and axis of bar graphs that the values provided are estimations only.

It was already explicitly stated in the text that these are estimations. We have now emphasized this in the text and Figure 7 as requested by the Reviewer.

Furthermore, a detailed procedure of how spiking threshold and integration of the number of inputs depending on their location on the soma or dendrites to trigger spiking should be added to the methods. Rough estimates based on summations of EPSP amplitudes at resting potential are providing a useful basis but they neglect the biophysical properties of motor neurons themselves regarding summation of EPSPs, which is by itself a complex problem requiring a fine analysis of the dynamics of EPSCs and biophysical properties of the postsynaptic targets.

The detailed procedure of how the spike threshold and the number of converging inputs was estimated is now provided in the text (pages 12-14)

The analysis provided in Figure 7 provides a conceptual framework showing that the different MNs required unequal excitatory drive for their recruitment. Future studies combining detailed biophysical and computational analyses will address the importance of the dynamics of the postsynaptic MNs.

Minor comments

- Figure 1 panel: indicate how V2a or MNs were labeled on the panel or at least in the legend- information is lacking to fully appreciate the data.

We now explain how V2a and MNs were labeled in the legend

- Figure 1. Panel a V2a label - is this a Z stack projection of a whole segment? if yes why are there only 9 cells visible, which is not matching the quantification given by the authors. Since the analysis is done per hemi segment it is critical to show a representative projection of the entire hemisegment for this region where the segment borders are highlighted and all labeled cells are shown.

We agree with the Reviewer and we have now changed the Z-stack projections to show almost one segment. The number of MNs and V2a INs are within the range shown in the graph of panel b.

- Legend Figure 1. Missing a word “Number of intersegmental V2a INs the 4 adjacent segments (2 rostral and 2 caudal) converging onto slow, intermediate and fast motor neurons”

This was corrected – thank you.

- Figure 2. Panel D - how many EPSPs were analysed for each speed module? Would be easier to see individual values plotted as a scatter rather than a bar chart. Also what is the failure rate and did the authors average traces in case of failure or not? Does every V2a AP result in an EPSP in the paired MN?

The amplitude of EPSPs elicited by single action potentials was measured from an average of all the sweeps (50 sweeps). This is now indicated in the Methods on page 24, paragraph 2.

We included all the sweeps in the analysis. Although the amplitude of individual EPSPs varied, they were elicited reliably without failure. We have now changed the traces in Figure 2 to include 10 sweeps and the average of all the 50 sweeps.

- Figure 3. Here the authors only show the relative % of synapses targeting the dendritic vs the somatic compartments, however the data can be further unpacked. For example it would be interesting to see what are the actual total numbers of V2a synapses on a given slow vs fast MN? Are there any differences? Also you could check the average no of synapses made by a bursting vs a non bursting V2a?

The main finding of this analysis is that the synaptic sites of the bursting- and non-bursting V2a INs are spatially segregated.

Our analysis allowed us to identify synaptic contact made by a single V2a IN onto a single MN. It is not possible to determine the total number of V2a synapses on a given MN because we are not able to label all the presynaptic V2a INs and a single MN.

Reviewer #3

1. Results related to convergence, Figure 1: Here, the authors have used retrograde labeling of MNs to count the number of MNs in a segment and relate this to the number of V2a INs either in that hemisegment or adjacent hemisegments. As V2a INs form electrotonic synapses with MNs, the dye is transported and double labelled V2a INs can be counted. They then demonstrate that there are fewer V2a INs than the number of MNs they innervate. If there is convergence of V2a INs onto MNs, then we should see more than one V2a IN labelled when a single MN is filled, for example. If every MN receives V2a input, then the data would support that there is divergence of V2a INs – that a single V2a IN innervates more than one MN. But do they support convergence – that more than one V2a IN projects to a single MN?

We have now added a new Supplementary Figure 1 showing data of dye-coupling when a single MNs were filled with neurobiotin. The results show that each MN is dye-coupled to 2-4 V2a INs.

What do the numbers in Fig 1e and Fig 1h mean – that is, how were they derived?

The numbers in these graphs represent the mean number of dye-coupled V2a INs labeled by injection of neurobiotin in slow, slow/intermediate and fast muscles in one myotome (hemisegment). The data are from 6 different animals in each condition. This is now clarified in the Methods and in the Figure Legends.

(On a minor related note, do 100% of V2a INs form electrotonic synapses with MNs? And are these both bursting and non-bursting types?) Convergence is a key point (see Fig 7), so should be made crystal clear.

Mixed electrical and chemical synapses were found in 65% of the dually recorded V2a IN-MN pairs, the rest were only connected via chemical synapses. There was no difference between bursting and non-bursting V2a INs. This is now clarified in the text (page 6, lines 3-5).

2. Plasticity. The authors use this term repeatedly, and it is unclear what they mean. If they are referring to non-linear summation of EPSPs mediated by NMDA receptors, I would not call it plasticity. As it stands, I think the term is a distraction at best, and does not add to the important findings of this manuscript. At minimum, their use of the term should be clearly defined and demonstrated.

We agree with the Reviewer and we have now refer to non-linear summation of EPSPs.

Minor comments:

1. Last sentence on p.4 has no predicate.

This was corrected – thank you.

2. p.6, top paragraph, last sentence (“In addition...”): the relevance/meaning of this sentence is opaque.

This sentence has been removed to avoid confusion.

3. p.7, last few lines: 82% of slow MNs have dendritic inputs, so this is most, whereas fast MNs have 25% somatic – this is not most.

This has been revised – thank you.

4. I'm not sure that Fig 2E contributes

This panel has been omitted.

5. p.10, middle – delay the fast module? Why a time delay here?

We did not mean a time delay, but to prevent the early recruitment of the fast module in favor of the slow one. The word “delay” was not appropriate and we have now changed to “postpone”.

6. Fig 6, amplitude graphs: it is not clear how the amplitude was calculated given the neurons are spiking – are the spikes included in the amplitude calculations? Is this reasonable? Is it meaningful?

The spikes were removed by smoothing the traces and only the amplitude of the underlying oscillation was measured. This is now clarified in the Methods (page 25, lines 11-17).

7. Top paragraph p.12 – it would be helpful if the number of neurons was reported throughout, not just in the first instance.

We agree with the Reviewer and the numbers are now reported throughout.

8. p.12 – 31 V2a INs is still a huge number considering the number in the hemisegment... the authors may want to remind the reader of this here.

The number of V2a INs are indicated again in this section as suggested by the Reviewer.

9. The authors refer to morphological features of V2a INs throughout – do they mean axonal projections? If other morphological features were quantified, they should be presented. Otherwise, the term should be clarified.

We only refer to the axonal projections and this has been clarified.

10. Given the dendritic non-linearities in MNs, even though they're shown to be NMDA receptor dependent, perhaps a mention of the extensive literature on dendritic non-linearities in mammalian MNs would be appropriate? Also, Brownstone et al 1994 showed non-linear increase in MN excitation during cat locomotion – this could be relevant?

We agree with the Reviewer – we now refer to the extensive literature on dendritic non-linearities in mammalian MNs.

11. SEM – In my view, there is little indication to use the SEM in biology, in which we're

interested in variability. For eg, in Fig 2D, it would be helpful to show scatter plots or box-whisker plots, or at minimum show and report the standard deviation rather than the SEM.

We agree with the Reviewer and we have now changed the graph to box-whisker plot.

But I do feel like I'm blowing against the wind when I raise this point in 9 out of 10 reviews at least!

The answer is blowing in the wind...

Reviewers' comments:

Reviewer #1 (Remarks to the Author):

The authors have revised their manuscript according to my criticisms and suggestions. I have no further suggestions.

Reviewer #2 (Remarks to the Author):

Regarding the response of authors to points raised, the authors provided multiple explanations and now only few but important points remain:

Major

1- High specificity of the Tg(Chx10:GFP) transgenic line used for identifying V2a in the study

The major concern with usage of this line, as many BAC-generated lines, is not the specificity of the Chx10 transcription factor but the fact that the expression of GFP does not strictly reflect the chx10 promoter.

The authors now specify that in the adult zebrafish we have never detected any expression of GFP in retrogradely labeled MNs and provide evidence for co-labeling with Chx10 antibody. Their answer on this important point is satisfactory.

2- Voltage clamp recording would constitute the proper confirmation of the model proposed by the authors based on pharmacology only.

3- Statistics need to be improved

a) Fig2. Page 7 lines 152-158 referring to Panel D: The authors now added p values from t-test to the results reported however here the authors are making 2 comparisons.

- Comparison (1) bursting cells across all groups result in higher amplitude EPSPs in MNs compared with non-bursting neurons.

- Comparison (2) bursting V2as from the slow module result in higher amplitude EPSPs compared with their counterparts from the intermediate and fast modules.

Both of these comparisons are based as far as it is reported on 3 separate t-tests for which the authors report only the p value.

Instead the authors should perform a 2-way anova in which they have 2 factors: bursting / non-bursting vs speed module identity (slow / intermediate / fast). Please report the full Anova values (F, df and p values) together with the values for the interactions across factors. Please advise.

If the authors decide to perform t-test(s) that they need to carry out all the possible combinations and do the appropriate post-hoc Bonferonni correction of $p = 0.05$ divided by the total number of tests. This approach however is not as rigorous to assess interactions across factors such as they claim, i.e. that slow bursting V2as result in higher amplitude EPSPs compared with the other cell types.

b) Fig. 4 & 5:

Need to improve statistical reporting and include full details: F values, df and p values together with Ns. This should be done for each level of the comparison carried out and for all interactions between factors.

c) Fig. 6:

Again, need to provide full info in terms of statistical analysis. What type of test was carried out? I would assume a T-test?

3 Showing diversity of cell responses and connectivity patterns:

a- In Fig 2C,D the authors show that fast MNs indeed receive input from bursting V2as. However, the authors later make the claim, based on anatomical data, that fast MNs receive preferential input from non-bursting V2a. Looking at the plots in panel D it looks like fast MNs actually receive equal numbers of inputs in either burst or single spike form. This is still confusing: Can the authors clarify whether in your classification a bursting cell can become non-bursting or the reverse? Or is it the case that single V2as can cycle through periods in which they have bursting vs single spike activity?

b- Did the authors observe a complete segregation of functional and anatomical profile of these 2 V2a classes? i.e. did the authors ever see a bursting V2a with a bifurcating axon?

c- Please provide full statistics for the correlations carried out in panels e and f - need to state test type, provide R value together with corresponding p-values.

d- Can the authors provide information about the APs generated by bursting vs non burstings V2a? I.e. do they have different resting membrane potentials, thresholds to spike, AP heights and widths? This is important to gauge whether there are any factors which might bias their recruitment and function.

e- There was a misunderstanding in terms of my previous request for this figure regarding the number of synapses. The SV2 staining on a given MN would allow the authors to count the total number of synapses present in the dendritic vs the somatic compartment. Given the claim that the authors can trace individual V2a axonal connections on a given MN the authors can show the precise number of synapses formed by a single given V2a rather than just report %. Please advise.

5. Reference to recently published work:

New publication by Bhumbra & Beato 2018 shows glutamatergic connections among MNs themselves. The authors's previous publication demonstrated electrical connections between MNs. Isn't it possible that fast MNs in particular will receive inputs from other non V2a- sources? See beautiful retrograde labeling from Arber's group showing a preponderance of dorsal located interneurons projecting onto motor neurons. This point and this reference is not addressed/cited in the manuscript and should be added to discussion.

Minor

Fig 1. Can the authors mark the borders of the spinal segment illustrated in all panels from this figure? Especially a-c as requested for revisions. This would help the reader better assess the relationship between the numbers stated in your bar plots vs the numbers of cells present in the picture.

Reviewer #3 (Remarks to the Author):

This is a resubmission of a manuscript that I have previously reviewed. The authors have significantly improved the manuscript, and have added new data and analysis.

I have but a few minor comments:

1. Since the first submission, a paper has come out from the Pfaff group indicating 2 types of V2a INs in mice, one that down-regulates Chx10 and projects up to the brain stem and the other that continues to express Chx10 and does not project to the brain stem. I would not want the authors to speculate beyond their level of comfort, but this reader for one wonders if the ones that project up might correspond to the non-bursting V2a INs here. It is indeed interesting that both groups have found a dichotomy but based on different factors.

2. Around line 400: "In contrast, the module encoding fast, more vigorous locomotor movements relies on a large number of converging non-bursting-type V2a interneurons and fast motor neurons, with less involvement of NMDA-dependent nonlinearity. The fast module relies primarily on a linear integration mode with a narrow time window for coincidence detection." This bit has me thinking about necessity and sufficiency... in particular, could it be that the fast module relies on inputs other than those from V2a INs?

3. Reviewer 2 has raised a number of points that could be addressed with voltage clamp experiments. While I agree that voltage clamp experiments would add understanding, it is my view that the manuscript sufficiently makes and supports a number of key findings even in the absence of voltage clamp recordings.

4. I am not sure why I continue to have a stumbling block about the ordinates of Figs 1e and 1h. Is this the number of V2a INs per MN or per MN pool? This should be in the label, it seems to me.

5. With respect to the response about point 6, the amplitude graphs: I am satisfied with the figure as it stands, but note that smoothing out action potentials – comprised of active conductances – is not a great way to "measure" membrane potential.

6. Finally, on p. 7, p. 15, and Figure 3 legend, the authors still talk about morphological features when really they mean axonal projections, as indicated in their response.

Reviewer #1:

The authors have revised their manuscript according to my criticisms and suggestions. I have no further suggestions.

We thank this Reviewer for the initial comments that helped improve the manuscript.

Reviewer #2:

Regarding the response of authors to points raised, the authors provided multiple explanations and now only few but important points remain:

We thank this Reviewer for all the detailed comments/criticisms that have improved the manuscript and the way the data are reported. We have now taken into account the additional comments as detailed below.

Major

1- High specificity of the Tg(Chx10:GFP) transgenic line used for identifying V2a in the study

The major concern with usage of this line, as many BAC-generated lines, is not the specificity of the Chx10 transcription factor but the fact that the expression of GFP does not strictly reflect the chx10 promoter.

The authors now specify that in the adult zebrafish we have never detected any expression of GFP in retrogradely labeled MNs and provide evidence for co-labeling with Chx10 antibody. Their answer on this important point is satisfactory.

Thank you for suggesting these experiments.

2- Voltage clamp recording would constitute the proper confirmation of the model proposed by the authors based on pharmacology only.

The lack of correlation between MN input resistance and the amplitude of the EPSPs argues for a difference in synaptic current between the different speed modules. While we agree that voltage-clamp experiments would have been informative, they do not seem crucial for the conclusions reached in this study.

3- Statistics need to be improved

a) Fig2. Page 7 lines 152-158 referring to Panel D: The authors now added p values from t-test to the results reported however here the authors are making 2 comparisons.

- Comparison (1) bursting cells across all groups result in higher amplitude EPSPs in MNs compared with non-bursting neurons.

- Comparison (2) bursting V2as from the slow module result in higher amplitude EPSPs compared with their counterparts from the intermediate and fast modules.

Both of these comparisons are based as far as it is reported on 3 separate t-tests for which the authors report only the p value.

Instead the authors should perform a 2-way anova in which they have 2 factors: bursting / non-bursting vs speed module identity (slow / intermediate / fast). Please report the full Anova values (F, df and p values) together with the values for the interactions across factors. Please advise.

If the authors decide to perform t-test(s) that they need to carry out all the possible combinations and do the appropriate post-hoc Bonferonni correction of $p = 0.05$ divided by the total number of tests. This approach however is not as rigorous to assess interactions across factors such as they claim, i.e. that slow bursting V2as result in higher amplitude EPSPs compared with the other cell types.

We have followed the Reviewer's advice and performed a two-way repeated-measures ANOVA. We now provide the full ANOVA values in the Fig. 2 legend and described the statistics in the text (page 7; lines 11-19).

b) Fig. 4 & 5:

Need to improve statistical reporting and include full details: F values, df and p values together with Ns. This should be done for each level of the comparison carried out and for all interactions between factors.

We have now included the full statistical details as requested by the Reviewer in figure legend.

c) Fig. 6:

Again, need to provide full info in terms of statistical analysis. What type of test was carried out? I would assume a T-test?

We have now included the full statistical details as requested by the Reviewer in figure legend.

3 Showing diversity of cell responses and connectivity patterns:

a- In Fig 2C,D the authors show that fast MNs indeed receive input from bursting V2as. However, the authors later make the claim, based on anatomical data, that fast MNs receive preferential input from non-bursting V2a. Looking at the plots in panel D it looks like fast MNs actually receive equal numbers of inputs in either burst or single spike form. This is still confusing: Can the authors clarify whether in your classification a bursting cell can become non-bursting or the reverse? Or is it the case that single V2as can cycle through periods in which they have bursting vs single spike activity?

During these experiments, it was much difficult to find fast bursting V2a INs than slow or intermediate ones. The graph in Fig 2D shows the different in the amplitude of the EPSPs induced by bursting- vs non-bursting-type V2a INs. Each data point is from a different experiment and it is likely that we recorded V2a INs with identical properties in different experiments. Therefore, the data points reflect the number of V2aIN recorded.

The bursting or non-bursting property of V2a INs was determined within 5 min after establishing whole-cell recordings. Non-bursting V2a INs could never become bursting even after a prolonged recording (>40 min). Most bursting V2a INs maintain their bursting property even after a prolonged recording (>40 min).

b- Did the authors observe a complete segregation of functional and anatomical profile of these 2 V2a classes? i.e. did the authors ever see a bursting V2a with a bifurcating axon?

None of the bursting V2a INs had a bifurcating axon. This is made clear in the text (page 7, last two sentences).

c- Please provide full statistics for the correlations carried out in panels e and f - need to state test type, provide R value together with corresponding p-values.

The test type and p values are now given in figure legend.

d- Can the authors provide information about the APs generated by bursting vs non burstings V2a? I.e. do they have different resting membrane potentials, thresholds to spike, AP heights and widths? This is important to gauge whether there are any factors which might bias their recruitment and function.

The factors influencing the recruitment of V2a INs are being examined together with the pattern of connectivity between V2a INs. It is premature to provide such information, which seems beyond the scope of the present manuscript.

e- There was a misunderstand in terms of my previous request for this figure regarding the number of synapses. The SV2 staining on a given MN would allow the authors to count the total number of synapses present in the dendritic vs the somatic compartment. Given the claim that the authors can trace individual V2a axonal connections on a given MN the authors can show the precise number of synapses formed by a single given V2a rather than just report %. Please advise.

The aim here was to examine the relative distribution of dendritic vs somatic synaptic contacts. Our results show that there is a differential distribution of these synaptic contacts on slow, intermediate and fast MNs. The analysis of the precise number of synapses would require EM serial reconstructions, which is beyond the scope of this study.

5. Reference to recently published work:

New publication by Bhumbra & Beato 2018 shows glutamatergic connections among MNs themselves. The authors's previous publication demonstrated electrical connections between MNs. Isn't it possible that fast MNs in particular will receive inputs from other non V2a- sources? See beautiful retrograde labeling from Arber's group showing a preponderance of dorsal located interneurons projecting onto motor neurons. This point and this reference is not addressed/cited in the manuscript and should be added to discussion.

We have now cited the paper by Bhumbra and Beato.

We appreciate that the Reviewer points this out, but we do not claim that MNs only receive input from V2a INs. However, ablation or silencing of V2a INs affected both fast and slow swimming movements (Eklöf-Ljunggren et al., 2012 PNAS; Sternberg et al. 2016 Current Biology).

Minor

Fig 1. Can the authors mark the borders of the spinal segment illustrated in all panels from this figure? Especially a-c as requested for revisions. This would help the reader better assess the relationship between the numbers stated in your bar plots vs the numbers of cells present in the picture.

We now indicate the exit of the ventral root, which give an indication of a border between two segments.

Reviewer #3:

This is a resubmission of a manuscript that I have previously reviewed. The authors have significantly improved the manuscript, and have added new data and analysis.

We thank this Reviewer for the constructive criticism and the comments that help improve the manuscript

I have but a few minor comments:

1. Since the first submission, a paper has come out from the Pfaff group indicating 2 types of V2a INs in mice, one that down-regulates Chx10 and projects up to the brain stem and the other that continues to express Chx10 and does not project to the brain stem. I would not want the authors to speculate beyond their level of comfort, but this reader for one wonders if the ones that project up might correspond to the non-bursting V2a INs here. It is indeed interesting that both groups have found a dichotomy but based on different factors.

It is interesting that two independent studies revealed two types of V2a INs in the zebrafish and mouse. It is premature to draw any definitive comparison of their firing properties and connectivity as this information is currently not available in mouse.

2. Around line 400: “In contrast, the module encoding fast, more vigorous locomotor movements relies on a large number of converging non-bursting-type V2a interneurons and fast motor neurons, with less involvement of NMDA-dependent nonlinearity. The fast module relies primarily on a linear integration mode with a narrow time window for coincidence detection.” This bit has me thinking about necessity and sufficiency... in particular, could it be that the fast module relies on inputs other than those from V2a INs?

Ablation of V2a INs affected swimming both at slow and fast speeds. Although our results do not exclude that MNs could also receive input from other INs, V2a INs represent a major source of excitation to MNs. Ablation or silencing of V2a INs affected both fast and slow swimming movements.

3. Reviewer 2 has raised a number of points that could be addressed with voltage clamp experiments. While I agree that voltage clamp experiments would add understanding, it is my view that the manuscript sufficiently makes and supports a number of key findings even in the absence of voltage clamp recordings.

We agree with the Reviewer and thank him/her for pointing this out.

4. I am not sure why I continue to have a stumbling block about the ordinates of Figs 1e and 1h. Is this the number of V2a INs per MN or per MN pool? This should be in the label, it seems to me.

We have labeled the Figs 1e and 1h to indicate in the x-axis corresponds to MN pools. This is also indicated in the corresponding figure legend.

5. With respect to the response about point 6, the amplitude graphs: I am satisfied with the figure as it stands, but note that smoothing out action potentials – comprised of active

conductances – is not a great way to “measure” membrane potential.

Thank you.

6. Finally, on p. 7, p. 15, and Figure 3 legend, the authors still talk about morphological features when really they mean axonal projections, as indicated in their response.

This has now been changed – thank you.

REVIEWERS' COMMENTS:

Reviewer #2 (Remarks to the Author):

**The authors satisfactorily answers every questions.
Great work !**

Reviewer #3 (Remarks to the Author):

The authors have revised the manuscript in response to reviews, and have done a stellar job. I have no further concerns.

I note that the colour-coding of the revisions based on the reviewer is special!

I suggest acceptance, with no need to turn around to take a final peek and see why it's so unique.